# VideoChat-Flash: Hierarchical Compression for Long-Context Video Modeling

Xinhao Li[2,1]*   Yi Wang[1]*†   Jiashuo Yu[1]*   Xiangyu Zeng[2,1]   Yuhan Zhu[2]
Haian Huang[1]   Jianfei Gao[1]   Kunchang Li[3]   Yinan He[1]   Chenting Wang[1]
Yu Qiao[1]   Yali Wang[3,1]   Limin Wang[2,1]†
[1]Shanghai AI Laboratory   [2]Nanjing University
[3]Shenzhen Institutes of Advanced Technology, Chinese Academy of Sciences
`https://github.com/OpenGVLab/VideoChat-Flash`

## Abstract

Long-context video modeling is critical for multimodal large language models (MLLMs), enabling them to process movies, online video streams, and so on. Despite its advances, handling long videos remains challenging due to the difficulty in efficiently understanding the extremely long video context. This paper aims to address this issue from aspects of the model architecture, training data, training strategy, and evaluation benchmark. First, we propose a novel **Hi**erarchical video token **Co**mpression (**HiCo**) method, which leverages visual redundancy in long videos to compress long video context from Clip-level to Video-level, reducing the computation significantly while preserving essential details, achieving an extreme compression ratio of approximately **1/50** with almost no performance loss. Second, we introduce a multi-stage **short-to-long learning** scheme, a large-scale dataset of real-world long videos named **LongVid**, and a challenging *"Multi-Hop Needle-In-A-Video-Haystack"* benchmark. Finally, we build a powerful video MLLM named **VideoChat-Flash**, which shows a leading performance on both mainstream long and short video benchmarks at the 2B and 7B model scales. It first gets **99.1%** accuracy over 10,000 frames in NIAH among open-source models.

## 1 Introduction

Long-context video modeling stands as one of the most crucial capabilities within multimodal large language models (MLLMs). This capability empowers MLLMs to proficiently manage hours-long movies, documentaries, and online video streams, all of which demand sophisticated long video processing. Recent advances in MLLMs (Wang et al., 2022; 2024e; Li et al., 2025; 2024c; Zhang et al., 2023a; Cheng et al., 2024; Zhang et al., 2024c; Lin et al., 2024; Xu et al., 2024; Li et al., 2024a; Bavishi et al., 2024; Li et al., 2023a) perform well in short video understanding. However, it remains challenging to build MLLMs for processing extremely long videos (lasting for hours or even longer). The difficulty lies in how to enable MLLMs to efficiently understand the extremely long video context brought by long videos.

Inspired by large language models (LLMs) with long context, modeling multimodal long context is widely studied from several perspectives. Some work (Reid et al., 2024; Xue et al., 2024) represented by Gemini-1.5-Pro (Reid et al., 2024) address it by training well-performed MLLMs on long-form corpus e.g. lengthy text and thousands of frames from videos, minimizing the gap between the evaluation and learning. Although the progress in system construction and hardware has made it possible to train and infer with super-long multimodal contexts, such super-long multimodal contexts have significantly reduced the training and inference efficiency of models. (For Gemini-1.5-Pro (Reid et al., 2024), a one-hour video will be converted into 921,600 tokens). Meanwhile, the high redundancy in long video context makes it particularly difficult for models to understand. Some previous efforts (Song et al., 2024; Li et al., 2024d; Shen et al., 2024) have been made to compress video tokens in order to achieve higher training and inference efficiency for long videos. However, the compression of visual content inevitably leads to the loss of detailed information. In

---

* Equal contribution. † Corresponding authors.

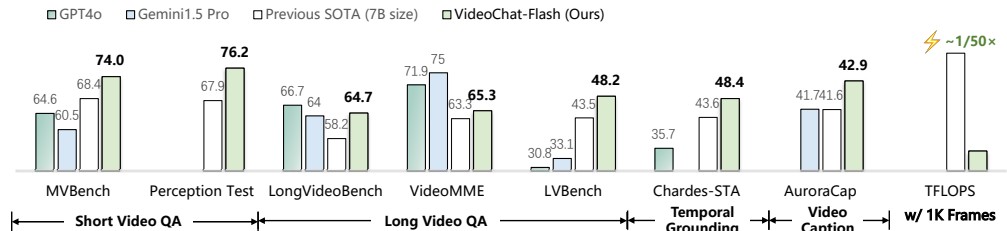

Figure 1: **Comparison results on various generic video-linguistic tasks.**

some long video understanding benchmarks, certain current long video models even perform worse than some image MLLMs. Therefore, how to *strike a balance between performance and efficiency* remains a significant challenge. In this paper, we attempts to address the above issues from the model architecture, training data, training strategy, and evaluation benchmark.

First, we propose a novel **Hi**erarchical video token **Co**mpression method (**HiCo**) to model the long video context efficiently, which defines the compression of the long video context into two stages. First, we segment the long video into multiple clips. Then, at the Clip-level, we utilize the spatio-temporal attention of the video encoder and the similar token merging to aggregate the key information between frames, thereby reducing the redundancy of inter-frame features. Subsequently, we take advantage of the sparsity of attention when the LLM processes long video tokens, further discard the video tokens that are irrelevant to the current task at the Video-level. HiCo could achieve an extreme compression ratio of approximately 1/50 with almost no performance loss. Additionally, we have conducted thorough explorations of other designs such as video sampling and timestamp awareness prompt.

Second, to further enrich the existing long video training corpus, we construct **LongVid**, a dataset that contains 114,228 long videos and 3,444,849 question-answering pairs. With LongVid, we have designed a multi-stage training strategy named **short-to-long learning**. The main idea is to first utilize image and short video data to learn basic visual perception abilities. Then, through the joint training of short video and long video data, the model is enabled to handle videos of different lengths and different types of tasks. In addition, we design a new evaluation benchmark named *"Multi-Hop Needle In A Video Haystack"*, which is more challenging and can better examine the model's complex reasoning abilities regarding long videos.

Finally, we develop a powerful video MLLM named **VideoChat-Flash**, as shown in fig. 1, which achieves remarkably leading performance with extremely high efficiency on various video understanding benchmarks. Even with a 7B size, it outperforms closed-source models such as GPT-4o (OpenAI, 2024) and Gemini-1.5-Pro (Reid et al., 2024).

## 2 RELATED WORK

**Multimodal Large Language Models for Video Understanding.** Recent advancements in multimodal large language models (MLLMs) have shown significant promise in video understanding. Most of them (Li et al., 2025; 2024c; Wang et al., 2024e; Lin et al., 2024; Zhang et al., 2023a; 2024c;d) focus on the understanding of minute-level videos, and some works (Reid et al., 2024; Song et al., 2024; Wang et al., 2024d; Shen et al., 2024; Xue et al., 2024; Shu et al., 2024; Huang et al., 2024) have further tried to handle longer hour-level videos. To address the challenge of processing long videos, researchers focus on two key strategies: (1) extending the context window of the LLM (Reid et al., 2024; Zhang et al., 2024a; Xue et al., 2024; Wei & Chen, 2024) and (2) compressing the video tokens (Li et al., 2024d; Fei et al., 2024; Weng et al., 2025; Tan et al., 2024; Song et al., 2024; Shu et al., 2024; Zeng et al., 2024). For context extension, although the approach of expanding the context window enable the possibility of long video understanding, it falls short of reducing the high computational burden and processing costs induced by long videos, thereby imposing limitations on its practical application. For token compression, Methods represented by Llama-Vid (Li et al., 2024d) use a highly compact representation while preserving key information. The high compression ratio makes it difficult for such methods to achieve excellent long video understanding performance,

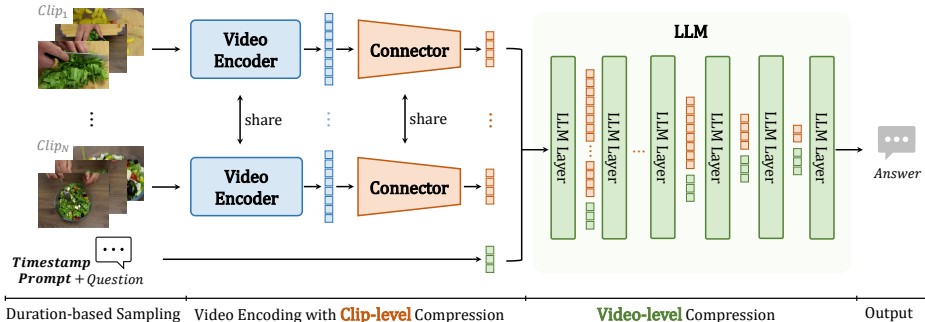

Figure 2: **Framework of VideoChat-Flash with Hierarchical Video Token Compression.** Video tokens will be compressed at the Clip-level by leveraging the local redundancy of the video modality during video encoding. Subsequently, during LLM processing, they will be compressed at the Video-level by taking advantage of the sparsity in the interaction between the text and the video.

and they may even be inferior to some MLLMs designed for image modeling. Therefore, how to build a Video MLLM that can balance both efficiency and performance remains a difficult challenge.

**Evaluation of Long Video Understanding.** In order to evaluate the ability of Video MLLMs to understand long videos, previous works (Song et al., 2024; Zhang et al., 2023b; Mangalam et al., 2023; Rawal et al., 2024; Wu et al., 2024; Zhou et al., 2024; Fu et al., 2024; Chandrasegaran et al., 2024; Wang et al., 2024c) have achieved this by collecting long videos and then designing various multiple-choice questions related to the content of these long videos. This approach is closer to real-world applications and can effectively examine the model's ability to understand and reason about long videos. However, when it comes to examining the model's capabilities for videos of different lengths, this method is not intuitive enough. Inspired by the popular "Needle in A Haystack" (NIAH) evaluation in long text context evaluation, some recent works (Zhang et al., 2024a; Zhao et al., 2024) have attempted NIAH for Video haystack. Nevertheless, it is difficult to assess complex reasoning abilities, and there may be information leakage. In this paper, we propose a more challenging *"Multi-Hop Needle-In-A-Video-Haystack"* is designed to address the above issues.

## 3 METHOD

### 3.1 HICO: EFFICIENT LONG VIDEO MODELING

To enable MLLMs to handle thousands of input frames, we propose a new video context compression paradigm named hierarchical compression (HiCo). This paradigm decomposes video context compression into two main stages: 1. **Clip-level compression for unimodal visual redundancy** during the encoding of long videos. 2. **Video-level compression for cross-modal visual redundancy** within the context interaction in the LLM. Based on this framework, we have designed an innovative efficient Video MLLM architecture, VideoChat-Flash, as illustrated in fig. 2. Below, we elaborate on our specific design details from data input to model output.

**Duration-based Sampling.** First, we need to perform frame sampling on the original video. Specifically, we sample a raw video with a duration of $D$ to obtain $T$ frames as input. Considering that the requirements for understanding short and long videos often differ, we aim to conduct dense sampling on short videos to capture detailed motions and sparse sampling on long videos to focus on event understanding. To this end, we have designed a Duration-based Sampling strategy:

$$T = \min(T_{\max}, \max(D, T_{\min})).$$ (1)

Simultaneously, we define the sampling density $\phi$ as follows:

$$\phi(T, D) = \frac{T}{D} = \frac{\min(T_{\max}, \max(D, T_{\min}))}{D}.$$ (2)

That is, for short videos where $D < T_{\min}$, $\phi = T_{\min}/D$, which increases as the video length decreases. For long videos where $D > T_{\max}$, $\phi = T_{\max}/D$, which decreases as the video length increases.

**Spatio-Temporal Compression Encoding for Clips.** Considering the substantial redundant and repetitive information, such as that of backgrounds and objects, present between adjacent frames in natural videos, we segment the original video frames into several equally sized clips. Each clip is then encoded using a ***Video Encoder with Spatio-Temporal Attention***, which effectively captures both key information and temporal redundancies within the clips, thereby significantly improving compression efficiency. We briefly offer some theoretical explanations here. First, we define the video features within a clip of length $T$ as $\mathbf{Y} = \{\mathbf{Y}_1, \mathbf{Y}_2, \cdots, \mathbf{Y}_T\}$, the compressed feature as $\mathbf{Z} = \mathcal{C}(\mathbf{Y})$, where $\mathcal{C}$ denotes a deterministic compression operation (e.g., average pooling). For commonly used Image Encoder (such as SigLIP (Zhai et al., 2023), etc.), they assume inter-frame independence, i.e., $p(\mathbf{Y}_1, \cdots, \mathbf{Y}_T) = \prod_{t=1}^{T} p(\mathbf{Y}_t)$. We can define the information loss caused by compression using the conditional entropy of the original features given the compressed feature:

$$L_c^{\text{img}} = H(\mathbf{Y}_1^{\text{img}}, \mathbf{Y}_2^{\text{img}}, \cdots, \mathbf{Y}_T^{\text{img}} | \mathbf{Z}) = \sum_{t=1}^{T} H(\mathbf{Y}_t^{\text{img}}) - H(\mathbf{Z}). \tag{3}$$

For the Video Encoder, it models the joint distribution $p(\mathbf{Y}_1, \cdots, \mathbf{Y}_T)$ using spatio-temporal attention, and its information loss can be expressed as:

$$L_c^{\text{vid}} = H(\mathbf{Y}_1^{\text{vid}}, \mathbf{Y}_2^{\text{vid}}, \cdots, \mathbf{Y}_T^{\text{vid}} | \mathbf{Z}) = H(\mathbf{Y}_1^{\text{vid}}, \mathbf{Y}_2^{\text{vid}} \cdots, \mathbf{Y}_T^{\text{vid}}) - H(\mathbf{Z})$$
$$= \sum_{t=1}^{T} [H(\mathbf{Y}_t^{\text{vid}}) - I(\mathbf{Y}_t^{\text{vid}}; \mathbf{Y}_1^{\text{vid}}, \mathbf{Y}_2^{\text{vid}}, \cdots, \mathbf{Y}_{t-1}^{\text{vid}})] - H(\mathbf{Z}), \tag{4}$$

where $I(\mathbf{Y}_t^{\text{vid}}; \mathbf{Y}_1^{\text{vid}}, \mathbf{Y}_2^{\text{vid}}, \cdots, \mathbf{Y}_{t-1}^{\text{vid}})$ represents the mutual information between the $t$-th frame and the previous $t-1$ frames, i.e., the redundant information between the $t$-th frame and its preceding frames. For most videos, it is evident that $I(\mathbf{Y}_t^{\text{vid}}; \mathbf{Y}_1^{\text{vid}}, \mathbf{Y}_2^{\text{vid}}, \cdots, \mathbf{Y}_{t-1}^{\text{vid}}) > 0$. Assuming the same $H(\mathbf{Z})$ and $\sum_{t=1}^{T} H(\mathbf{Y}_t^{\text{img}}) \approx \sum_{t=1}^{T} H(\mathbf{Y}_t^{\text{vid}})$, we thus have $L_c^{\text{img}} > L_c^{\text{vid}}$. A detailed proof is provided in the Appendix. In terms of specific implementation, we use ToMe (Bolya et al., 2022) as $\mathcal{C}$. Benefiting from the effectiveness of the video encoder in modeling spatio-temporal interactions, we achieve an extremely heavy compression while well retaining the key information, with each video frame being compressed to an average of only **16** tokens.

The compressed features from different clips are chronologically merged to form the final visual context. This representation is then aligned with the feature space of the LLM through a video-language projection. Furthermore, to reduce the cost of timestamp encoding, unlike previous approaches (Ren et al., 2024; Chen et al., 2024e) that depend on auxiliary modules or insert textual annotations into every frame, which introduces substantial computational overhead when processing long videos, we introduce a ***lightweight timestamp prompt*** appended after the video context: *"The video lasts for N seconds, and T frames are uniformly sampled from it."* We find that this straightforward approach is sufficient to enable the model to perceive the timestamps of the input video, achieving excellent performance on timestamp sensitive tasks such as temporal grounding (see table 1). Finally, the entire video context $\mathbf{X_v}$ can be represented as:

$$\mathbf{X_v} = \text{Concat}(\mathcal{F}(\mathbf{Z}_1), \mathcal{F}(\mathbf{Z}_2), \cdots, \mathcal{F}(\mathbf{Z}_{N_c}), \mathbf{X}_{\text{timestamp}}), \tag{5}$$

where $N_c$ is the number of clips, $\mathcal{F}$ is a MLP projection serving as a video-language connector.

**Progressive Visual Dropout in LLM.** Although clip-level compression has been carried out before, due to the possibility of longer-range visual redundancies in long videos (e.g. surveillance videos), and when an LLM responds to specific instructions regarding the visual input, it may not be necessary to continuously focus on the entire long video context. We consider conducting further video-level compression during the LLM inference stage. Recent works (Chen et al., 2025; 2024b) have explored acceleration strategies for MLLMs when processing short visual contexts. Most of them drop visual tokens based on the correlation between text tokens and visual tokens. In contrast, we find that when the LLM processes a long video context, it pays attention to the entire long video context at the shallow layers of the LLM, while focusing on the details of certain local moments at the deep layers (see the Appendix for specific visualizations). Based on this observation, we have designed a progressive visual dropout strategy, which is divided into two stages. At the shallow layers of the LLM, we uniformly drop a small number of video tokens (i.e. ***uniform drop***), reducing the

computation while maintaining the original spatio-temporal structure of the video context. At the deep layers of the LLM, we rely on the correlation between text tokens and video tokens to retain the most critical relevant information (i.e. **text-guided select**). We have found that this operation not only effectively improves the computational efficiency of the model but also slightly enhances the understanding performance of the model by reducing irrelevant visual noise.

## 3.2 LARGE-SCALE CORPUS FOR LONG VIDEO TRAINING

One of the challenges in long video model training is the shortage of large-scale, high-quality data. Though recent advances have mitigated this by long-form datasets of video-text pairs, these lack the instruction-following paradigm, such as (video, instruction, answer) triplets, crucial for multimodal reasoning. To address this, we introduce a large-scale long video instruction-tuning dataset named **LongVid**. It comprises 114,228 long videos (with an average duration of 367.3 seconds) and 3,444,849 question-answering (QA) pairs, covering five distinct task types: *long video captioning, temporal grounding, event relation recognition, scene relation recognition, and video event counting*. LongVid significantly surpasses previous datasets in both scale and average video length, enabling models to tackle a wide range of long video scenarios.

To construct the LongVid dataset, we follow three core steps: **(1)** first, for data source selection, we leverage diverse existing long video datasets that include Ego4D (Grauman et al., 2022), HowTo100M (Miech et al., 2019), HD-VILA (Xue et al., 2022), and MiraData (Ju et al., 2024); these datasets collectively cover multiple video types (e.g., movies, egocentric videos, news, interviews, how-to videos, and other in-the-wild long videos) to ensure the dataset's diversity. **(2)** Second, for event label curation, a key step that involves generating dense event labels for each long video, we first utilize high-quality short video captions tailored to each source dataset (e.g., Panda-70M (Chen et al., 2024c) for HD-VILA, CosMo (Wang et al., 2024a) for HowTo100M, Ego4D-HCap (Islam et al., 2024) for Ego4D, and the original captions for MiraData), then filter consecutive short video segments that can be reorganized into a single long video sequence, and subsequently construct timestamped event label sequences for each long video based on the aforementioned captions; specifically, for datasets with pre-existing high-quality event-level annotations (e.g., HT-Step (Afouras et al., 2024) for HowTo100M, Ego4D-HCap for Ego4D), we directly leverage these annotations as event labels, while for datasets lacking such annotations, we extract core events from the captions using a large language model (LLM). **(3)** Finally, for QA pair construction, we build multiple types of long video question-answering (QA) pairs using the video captions, event labels, and timestamps of the short video segments, with further details about the entire construction process provided in the Appendix.

## 3.3 MULTI-STAGE SHORT-TO-LONG LEARNING

Unlike studies (Zhang et al., 2024a; Xue et al., 2024) that use long-form text to extend the context window, we prefer that direct training on long-form videos minimizes the gap between training and testing, leading to better downstream evaluations. The training data are detailed in the Appendix.

**Stage-1: Video-Language Alignment.**   In this stage, we freeze the visual encoder and the large language model while training the compressor and the MLP to align the language with the compressed visual features. We use 0.5 million image-text pairs and 0.5 million short video-text pairs, and sample 4 frames from each video in training.

**Stage-2: Short Video Pre-training.**   To enhance the model's understanding of visual concepts, we conduct visual pre-training using 3.5 million images and 2.5 million short video-text pairs. During this stage, we sample 8 frames from each video.

**Stage-3: Joint Short & Long Video Instruction Tuning.**   To enable the model to handle a wide variety of video tasks, we collect 3.5 million instruction fine-tuning samples, including 1.1M images, 1.7M short videos (under 60 seconds), and 0.7M long videos (60∼3600 seconds). We mix the short and long video data to ensure the model retains fine-grained understanding while expanding its comprehension of long videos. The sampling method used is the duration-based sampling described in section 3.1, with the number of sampled video frames ranging from 64 to 512.

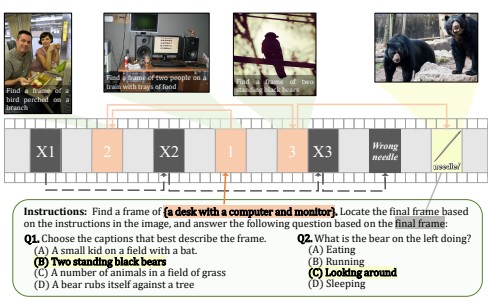

Figure 3: **An example of our Multi-Hop Needle in a Video Haystack**. The right path (1, 2, 3) is for finding the needle while the wrong path (X1, X2, X3) is for distraction. MLLMs are asked to both find the needle (Q1) and answer its related question (Q2).

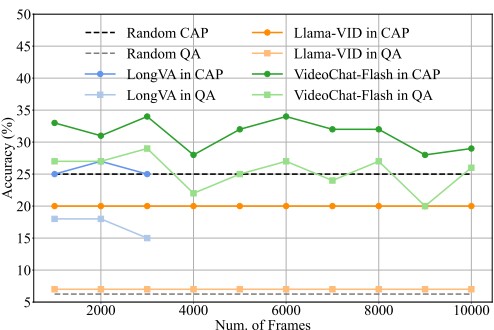

Figure 4: **Results on the Multi-Hop NIAH with 10,000 frames.** "CAP" represents the accuracy of finding the correct needle, while "QA" denotes the accuracy of answering the questions related to the correct needle while also finding the correct needle.

**Stage-4: Efficient Video Encoding Enhancement.** To enable the model to perceive higher resolutions, we employ an efficient post-finetuning strategy to adapt the original low-resolution video encoder to higher-resolution inputs. Specifically, we increase the input resolution of the video encoder from 224 to 448, freeze the LLM, and directly utilize 25% of the stage-3 data for post-finetuning the video encoding. We find that this simple, full-data strategy effectively enhances the video encoder's adaptability to higher-resolution video inputs.

## 3.4 MULTI-HOP NEEDLE IN A VIDEO HAYSTACK

Previous works (Zhang et al., 2024a; Xue et al., 2024) utilize the "Needle in a Video Haystack" (NIAH-Video) to evaluate the long video context understanding ability of models. Specifically, an image (commonly referred to as the "needle") was inserted into a long video and then the model under test was required to input the entire video and answer questions related to the needle. NIAH-Video assesses the model's capability to retrieve information from long videos. However, it has several drawbacks. Firstly, it is difficult to prevent images and questions similar to the needle from appearing in the model's training data, which leads to information leakage. Secondly, merely examining the model's visual retrieval ability is insufficient and lacks discrimination for evaluating its long video context understanding ability (many models can achieve an accuracy rate over 99%). There is a need to further evaluate its reasoning ability regarding the content.

To address the above issues, we have designed a new evaluation task called "Multi-Hop Needle in a Video Haystack" (MH-NIAH-Video). As shown in fig. 3, we insert a reasoning path composed of multiple images into the video haystack. Each image in this path has a randomly inserted position and the corresponding textual clues to help find the next image. Given the starting point of the reasoning path, the model needs to follow this path to find the needle and answer questions related to it. What's more, to prevent the model from skipping the step of finding the needle by relying on information leakage or memorizing the content of all images, we insert multiple wrong reasoning paths simultaneously while inserting the correct reasoning path. The model needs to find the correct needle (Q1) along the correct reasoning path based on the given starting point and then answer questions related to the needle (Q2). In a way, our multi-hop approach offers a much more robust evaluation of the long context understanding ability in Multimodal Large Language Models (MLLMs) compared to the previous NIAH-Video. In practice, all images are sourced from MS-COCO (Lin et al., 2014), making use of its human-annotated captions and question-answer pairs. It should be noted that even if the model can perfectly remember the content of MS-COCO, it will not be of much help in finding the needle, significantly reducing the likelihood of successful "cheating".

Table 1: **Results on comprehensive video-linguistic benchmarks.**

| Model | Size | Avg tokens per frame | MVBench | PerceptionTest | LongVideoBench | MLVU | VideoMME (*w/o & w sub.*) | | LVBench | Charades-STA | AuroraCap |
|---|---|---|---|---|---|---|---|---|---|---|---|
| | | | Avg | Val | Val | M-Avg | Overall | Long | Avg | mIoU | Avg |
| Avg. Duration | | | 16s | 23s | 473s | 651s | 1010s | 2386s | 4101s | 30s | 28s |
| *Proprietary Models* | | | | | | | | | | | |
| GPT-4V (OpenAI, 2023) | - | - | 43.7 | - | 59.1 | 49.2 | 59.9/63.3 | 53.5/56.9 | - | - | - |
| GPT-4o (OpenAI, 2024) | - | - | 64.6 | - | 66.7 | 64.6 | 71.9/77.2 | 65.3/72.1 | 30.8 | 35.7 | - |
| Gemini-1.5-Pro (Reid et al., 2024) | - | - | 60.5 | - | 64.0 | - | 75.0/81.3 | 67.4/77.4 | 33.1 | - | 41.7 |
| *Small Size MLLMs* | | | | | | | | | | | |
| Qwen2-VL (Wang et al., 2024b) | 2B | 1924 | 63.2 | - | - | - | 55.6/60.4 | - | - | - | - |
| InternVL2.5 (Chen et al., 2024d) | 2B | 256 | 68.8 | - | 46.0 | 61.4 | 51.9/54.1 | - | - | - | - |
| **VideoChat-Flash @448** | 2B | 16 | 70.0 | 70.5 | 58.3 | 65.7 | 57.0/63.9 | 44.9/54.0 | 42.9 | 45.2 | - |
| *Open-Source MLLMs* | | | | | | | | | | | |
| VideoChat2-HD (Li et al., 2024c) | 7B | 72 | 62.3 | - | - | 47.9 | 45.3/55.7 | 39.8/53.9 | - | 3.4 | - |
| InternVideo2-HD (Wang et al., 2024e) | 7B | 72 | 67.2 | 63.4 | - | - | 49.4/ - | - | - | - | - |
| LLaVA-OneVision (Li et al., 2024a) | 7B | 196 | 56.7 | 57.1 | 56.3 | 64.7 | 58.2/61.5 | - | - | 13.5 | 37.5 |
| LLaVA-OneVision (Li et al., 2024a) | 72B | 196 | 59.4 | 66.9 | 61.3 | 68.0 | 66.2/69.5 | - | - | - | - |
| LLaVA-Video (Zhang et al., 2024d) | 7B | 676 | 58.6 | 67.9 | 58.2 | 70.8 | 63.3/69.7 | - | - | - | 39.0 |
| VITA1.5 (Fu et al., 2025) | 7B | 256 | 56.8 | - | - | - | 56.8/59.5 | - | - | - | - |
| InternVL2 (Chen et al., 2024e) | 8B | 256 | 65.8 | - | 54.6 | 64.0 | 54.0/56.9 | - | - | - | 37.7 |
| InternVL2 (Chen et al., 2024e) | 76B | 256 | 69.6 | - | 61.1 | 69.9 | 61.2/62.8 | - | - | - | - |
| InternVL2.5 (Chen et al., 2024d) | 8B | 256 | 72.0 | - | 60.0 | 68.9 | 64.2/66.9 | - | - | - | - |
| Qwen2-VL (Wang et al., 2024b) | 7B | 1924 | 67.0 | 66.9 | - | - | 63.3/69.0 | - | - | - | 41.6 |
| Qwen2.5-VL (Bai et al., 2025) | 7B | 1924 | 69.6 | - | 56.0 | 70.2 | 65.1/71.6 | - | 45.3 | 43.6 | - |
| *Open-Source Long Video MLLMs* | | | | | | | | | | | |
| LLaMA-VID (Li et al., 2024d) | 7B | 2 | 41.9 | 44.6 | - | 33.2 | 25.9/ - | - | 23.9 | - | 30.9 |
| LongVU (Shen et al., 2024) | 7B | 64 | 66.9 | - | - | 65.4 | - /60.6 | - /59.5 | - | - | - |
| LongVA (Zhang et al., 2024a) | 7B | 144 | - | - | - | 56.3 | 52.6/54.3 | 46.2/47.6 | - | - | 34.5 |
| LongVILA (Xue et al., 2024) | 7B | 196 | 67.1 | 58.1 | 57.1 | - | 60.1/65.6 | 47.0/52.1 | - | - | - |
| Kangaroo (Liu et al., 2024) | 8B | 256 | 61.0 | - | 54.8 | 61.0 | 56.0/ 57.6 | 46.7 / 49.3 | 39.4 | - | - |
| **VideoChat-Flash @224** | 7B | 16 | 73.2 | 75.6 | 64.2 | 74.5 | 64.0/69.4 | 53.6/61.9 | 47.2 | **48.4** | - |
| **VideoChat-Flash @448** | 7B | 16 | **74.0** | **76.2** | **64.7** | **74.7** | **65.3/69.7** | **55.4/63.3** | **48.2** | 48.0 | 42.9 |

## 4 EXPERIMENTS

**Implementation details.** We employ UMT-L (Li et al., 2023b), token merging with MLP, and Qwen2-7B as visual encoder, connector, and LLM, respectively. When processing a long video, we divide it into shorter clips, each consisting of 4 frames. Each clip is compressed into 64 tokens, meaning that, on average, each frame is represented by 16 tokens. Regarding video-level compression, while we found that enabling it during training would slightly impair performance, so we only employ it during inference. We use only one-fourth of the full dataset for ablation. See Appendix for details.

### 4.1 GENERAL VIDEO UNDERSTANDING EVALUATION

**Leading performance.** We evaluate our model on six general video understanding benchmarks in question-answering format, including two short video benchmarks: MVBench (Li et al., 2024c) and Perception Test (Patraucean et al., 2024), and three long video benchmarks: LongVideoBench (Wu et al., 2024), MLVU (Zhou et al., 2024) and LVBench (Wang et al., 2024c), and a comprehensive benchmark, VideoMME (Fu et al., 2024), covering videos ranging from minute-level to hour-level durations. We further evaluate the temporal grounding and video caption tasks, using the Charades-STA (Gao et al., 2017) and AuroraCap (Islam et al., 2024). As depicted in table 1, our VideoChat-Flash achieves the best results on diverse VideoQA benchmarks within the 2B and 7B size category, significantly outperforming other approaches. Remarkably, its performance even eclipses that of models with substantially larger scales, such as InternVL2-76B, as well as proprietary models like GPT-4o and Gemini-1.5-Pro. Even when merely supplying timestamp information via a text prompt, our model has achieved remarkable performance in temporal grounding. Meanwhile, it also significantly outperforms other models in the video captioning task, even surpassing the proprietary GPT-4o and Gemini-1.5 Pro. This demonstrates the effectiveness of the comprehensive design of our model, data, and training strategies.

### 4.2 LONG VIDEO CONTEXT EVALUATION

**Baseline.** LongVA (Zhang et al., 2024a) and LLama-VID (Li et al., 2024d) are used as baselines. LongVA trains MLLMs using long text data, transfering the long context of LLM from text to video. LLama-VID accomplishes efficient inference of long videos by compressing each frame to only two tokens. Our model benefits from these two, so we take them as baselines.

**Single-Hop NIAH.** As shown in fig. 5, we follow the protocols in LongVA (Zhang et al., 2024a) for Single-Hop NIAH, we source a long video and sample frames uniformly from it. Then we add needles (indicating images) into the sampled image sequence at different positions. MLLMs are

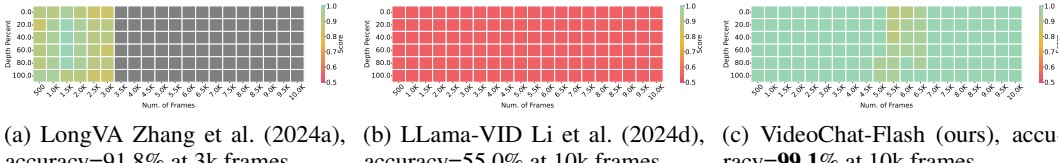

(a) LongVA Zhang et al. (2024a), accuracy=91.8% at 3k frames.

(b) LLama-VID Li et al. (2024d), accuracy=55.0% at 10k frames.

(c) VideoChat-Flash (ours), accuracy=**99.1**% at 10k frames.

Figure 5: **Results on the Single-Hop NIAH evaluation with 10,000 frames.**

fed with this long image sequence and answer the corresponding questions to the indicating images. We evaluate all models over 10,000 frames. Note our VideoChat-Flash delivers a 99.1% success rate in accurately retrieving the correct indicating image and answering the related question even across 10,000 frames. In comparison, LongVA gives a decent result close to 92% within 3000 frames while LLama-VID only achieves 55% accuracy. It demonstrates VideoChat-Flash's state-of-the-art performance in long multimodal context modeling.

Table 2: **Performance comparison on MH-NIAH benchmark**. Evaluated under 1000-frame setting with 266k input tokens. "Cap Score" denotes captioning score, and "QA Score" denotes question answering score.

| Model | Thinking | Token Per Frame | Cap Score | QA Score |
|---|---|---|---|---|
| Random | - | - | 25% | 6.25% |
| LlamaVid | × | 2 | 20% | 7% |
| LongVA | × | 144 | 25% | 18% |
| **VideoChat-Flash (Ours)** | × | 16 | 33% | 27% |
| Gemini2.5 Flash | × | 258 | 35% | 31% |
| | ✓ | 258 | 60% | 54% |

**Multi-Hop NIAH.** In this evaluations, MLLMs need to trace along the chain of indicating images, locate the needle, and answer its question. Two metrics "CAP" and "QA" are used to denote the accuracy of finding the correct needle and the accuracy of answering the questions related to the needle as well as finding the needle successfully, respectively. As shown in fig. 4, our VideoChat-Flash still beats all baselines. Specifically, VideoChat-Flash gives 31.3% and 25.4% in "CAP" and "QA" on average, higher than LongVA by around 8 points. It can be seen that compared with the single-hop NIAH, the multi-Hop NIAH presents a much more difficult challenge, which can better reflect the real gap between the capabilities of different models. As shown in table 2, we also have supplemented the results of Gemini 2.5 Flash and Gemini 2.5 Flash thinking on our MH-NIAH benchmark. Due to budget constraints, we only evaluated its performance under the 1000-frame setting, with an input token count of approximately 266k. Surprisingly, without enabling the reasoning mode, Gemini 2.5 Flash achieved a score only slightly higher than that of our VideoChat-Flash in the Multi-Hop NIAH. However, its score significantly improved when the thinking mode was activated. This further validates the fact that *the Multi-Hop NIAH task we designed truly requires more complex video reasoning capabilities rather than mere retrieval abilities for successful completion*.

## 4.3 ABLATION & ANALYSIS

**Effect of compression ratio.** As shown in fig. 6a, a lower compression ratio (i.e., fewer encoded visual tokens per frame on average) confers a substantial efficiency advantage for both short and long video inputs, and this advantage becomes more pronounced as the input length increases. Notably, as shown in fig. 6b, under our well-designed compression strategy, the detrimental impact of compression on video task performance is less significant than anticipated. For both short-video and long-video understanding tasks, appropriate compression can even enhance task performance; even under an extreme compression ratio of 2%, the model can still retain 95% of its performance. This finding gives us confidence in developing state-of-the-art video understanding models with low compression ratios.

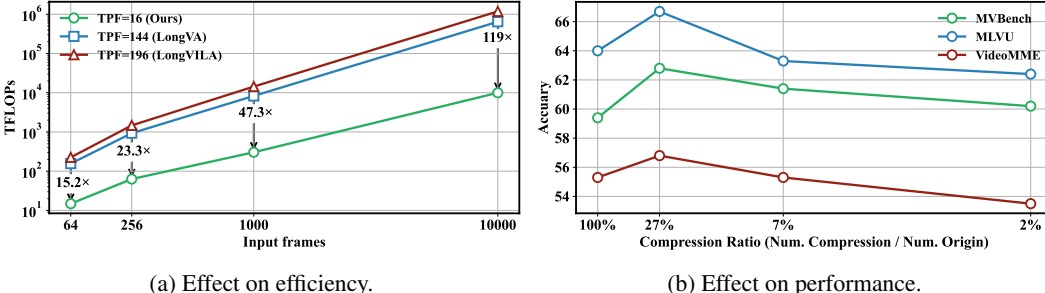

(a) Effect on efficiency.                    (b) Effect on performance.

Figure 6: **Effect of compression ratio.**

Table 4: **Effectiveness of video encoder.** We use the popular image encoder (SigLIP (Zhai et al., 2023)) and video encoder (UMT (Li et al., 2023b)) for comparison.

| Visual Encoder | FLOPs (G) | Latency (ms) | MVBench Avg | MLVU M-Avg | VideoMME Overall |
|---|---|---|---|---|---|
| *#tokens per frame=16, training data size=2M* | | | | | |
| SigLIP$_{SO400M}$@384 | 2679 | 79.7 | 60.2 | 62.0 | **53.5** |
| UMT-L@224 | 596 | 11.8 | 61.1(+0.9) | 60.0(-2.0) | 53.2(-0.3) |
| *#tokens per frame=16, training data size=8M* | | | | | |
| SigLIP$_{SO400M}$@384 | 2679 | 79.7 | 71.2 | 70.8 | 62.4 |
| UMT-L@224 | 596 | 11.8 | 73.5(+2.3) | 73.7(+2.9) | 62.7(+0.3) |

Table 5: **Effectiveness of visual dropout.** The Qwen2-7B we used has a total of 28 layers. "Uni." and "Attn." represent uniform drop and attention select respectively.

| Drop type/keep ratio | Drop layer | FLOPs (G) | Latency (s) | MLVU M-Avg | VideoMME Overall |
|---|---|---|---|---|---|
| - | - | 341.4 | 2.6 | 71.8 | 61.2 |
| Uni./0.5 | 4 | 242.8 | 1.9 | **71.2** | 60.4 |
| Attn./0.5 | 4 | 242.8 | 1.9 | 70.7 | **60.8** |
| Uni./0.5 | 18 | 295.2 | 2.2 | 71.7 | **61.8** |
| Attn./0.5 | 18 | 295.2 | 2.2 | **72.1**(+0.3) | 61.7(+0.5) |
| Attn./0.75,Attn./0.25 | 4,18 | 245.8 | 1.9 | 71.4 | 60.9 |
| Uni./0.75,Attn./0.25 | 4,18 | 245.8 | 1.9 | **72.0**(+0.2) | **61.1**(-0.1) |

**Ablation of various designs.** As shown in table 3, we have conducted comprehensive ablation studies on each design. In terms of the model, it can be observed that HiCo significantly reduces the computation (from 196 to 16 tokens per frame) while barely compromising the performance. Meanwhile, duration-based sampling and timestamp prompts play crucial roles in enhancing the performance. The further leap in performance mainly stems from the training strategy in short-to-long learning and a better mixture of training data.

Table 3: **Ablation of various designs** on data, model, and resolution. The baseline employs SigLiP-so400M (Zhai et al., 2023) as the vision encoder and Spatial donwsampling (196 tokens per frame) as the connector. It adopts a two-stage training strateay with image and short video following LLaVA (Liu et al., 2023).

| Settings | MVB Avg | MLVU M-Avg | VMME Overall | Charades mIoU |
|---|---|---|---|---|
| Baseline | 60.2 | 63.7 | 52.8 | |
| + HiCo | 61.1 | 60.6 | 53.2 | - |
| + short video pretraining | 66.5 | 62.4 | 53.9 | - |
| + duration-based sampling | 67.0 | 64.5 | 55.5 | - |
| + LongVid data | 66.5 | 68.3 | 55.8 | - |
| + Joint short & long sft | 73.2 | 74.5 | 64.0 | 48.4 |
| + High-res post ft | 74.0 | 74.7 | 65.3 | 48.0 |
| − timestamp prompt | 73.4 | 73.2 | 63.4 | 44.2 |

**Effectiveness of spatio-temporal compression encoding.** As shown in table 4, we have tested the most popular image encoder, SigLIP (Zhai et al., 2023), and the short video encoder, UMT (Li et al., 2023b), for encoding clips with heavy compression. We found that even when the computational cost is significantly lower, UMT can still achieve better performance on the short video task MVBench. Moreover, as the size of the training data increases from 2 million to 8 million, UMT outperforms SigLIP distinctly across various benchmarks. We believe that this is attributed to the spatio-temporal attention employed by UMT, which can aggregate the key information from different frames within a clip, thus enabling the learning of more compact compression features.

**Effectiveness of progressive visual dropout.** As shown in the table 5, at the shallow layers of the LLM, uniform dropout performs better than attention select on long video tasks. However, at the deep layers of the LLM, attention select shows better performance. Performing visual dropout at the deep layers can not only improve the computational efficiency but also enhance the performance. Combining uniform dropout and attention select can achieve a good balance between performance and efficiency. More relevant analyses and comparative experiments can be found in the Appendix.

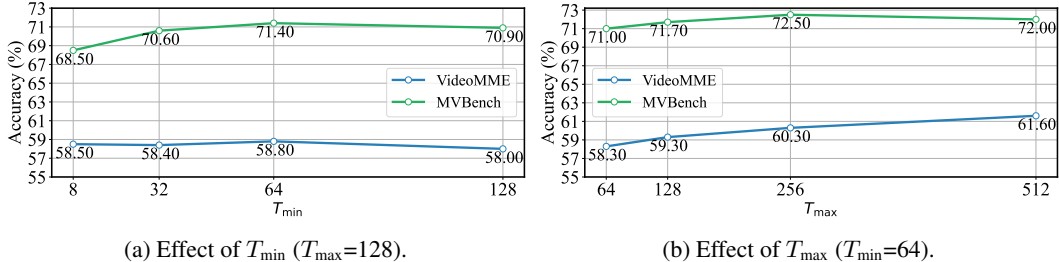

(a) Effect of $T_{\min}$ ($T_{\max}$=128).          (b) Effect of $T_{\max}$ ($T_{\min}$=64).

Figure 7: **Ablation of duration-based sampling.**

**Effectiveness of duration-based sampling.** As shown in fig. 7, A relatively large $T_{\min}$ (64) enables the model to better learn to model the fine actions and rapid movements in short videos during training, thereby enhancing the performance of short video understanding. Increasing $T_{\max}$ from 64 to 256 leads to a stable improvement in the understanding performance of both short and long videos. This indicates that more sampled frames can extract more accurate information from our long video data. When $T_{\max}$ reaches 512, there is a slight decline in the performance of short videos. Overall, it achieves a balance between the performance of short and long videos.

Table 6: **Ablation results of Progressive Visual Dropout in training/inference stages.**

| Training | Evaluation | VideoMME | MVBench |
|---|---|---|---|
| No Dropout | No Dropout | 53.2 | 61.1 |
| Progressive Visual Dropout | No Dropout | 53.6(+0.4) | 58.4(-2.7) |
| | Progressive Visual Dropout | 52.2(-1.0) | 57.9(-3.3) |

**Impact of progressive visual dropout on training.** As shown in table 6, ablation studies verify that the train-inference mismatch of Visual Drop (training with dropout vs. inference without) does not lead to performance degradation. Training-stage Visual Drop yields no significant performance gains and incurs slight performance attenuation, which we ascribe to its inherent mechanism of identifying and retaining key visual tokens via text-guided attention. We therefore apply Visual Drop exclusively during the inference phase based on these empirical results.

Table 7: **Results with different video encoder.**

| Video encoder | MVBench Avg | PerceptionTest Val | LongVideoBench Val | MLVU M-Avg | VideoMME (*w/o sub.*) Overall | LVBench Avg |
|---|---|---|---|---|---|---|
| Avg. Duration | 16s | 23s | 473s | 651s | 1010s | 4101s |
| UMT-L | 73.2 | 75.6 | 64.2 | 74.5 | 64.0 | 48.4 |
| InternVideo2-1B | 74.3(+1.1) | 76.3(+0.7) | 64.5(+0.3) | 73.4(-1.1) | 65.2(+1.2) | 48.7(+0.3) |

**Results with InternVideo2.** As shown in table 7, We also attempted to use the more powerful InternVideo2-1B (Wang et al., 2024e) as the video encoder. We found that a stronger video encoder can lead to better compressed representations.

## 5 CONCLUSIONS

In this paper, we address the challenge of long-context video modeling in MLLMs from the model architecture, training data, training strategy and evaluation benchmark. We design an efficient architecture for video MLLMs by introducing a hierarchical long video context compression method, which achieves an extreme compression ratio with nearly no performance loss. Regarding data and training, we propose a new long video training corpus and short-to-long learning strategy, which effectively enhances the model's understanding ability for videos of various lengths. Additionally, we developed a new and more challenging long video context evaluation benchmark. Our model demonstrated outstanding performance on various video understanding benchmarks, which validates the effectiveness of our proposed methods.

## 6 REPRODUCIBILITY STATEMENT

We provide all the necessary details to reproduce our experiments in section 4 and Appendix.

## 7 ACKNOWLEDGEMENTS

This work is supported by the National Key R&D Program of China (No. 2022ZD0160900), the Basic Research Program of Jiangsu (No. BK20250009), the Shanghai Artificial Intelligence Laboratory and the Collaborative Innovation Center of Novel Software Technology and Industrialization.

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

## A APPENDIX

In this appendix, we provide more details from the following aspects:

- § A.1: Statement on the Use of LLM.
- § A.2: Supplementary proof of Clip-Compression.
- § A.3: More Results & Discussions of VideoChat-Flash.
- § A.4: Implementation Details of VideoChat-Flash.
- § A.5: Dataset Details of LongVid.
- § A.6: Qualitative Results of VideoChat-Flash.

### A.1 STATEMENT ON THE USE OF LLM IN WRITING

LLM was used for polishing and review during the writing of this article.

### A.2 SUPPLEMENTARY PROOF OF CLIP-COMPREESION

To complete the derivation of section 3.1, we elaborate on the key steps using fundamental information-theoretic properties, including conditional entropy, joint entropy, and mutual information.

**Step 1: Conditional Entropy for Compression Loss**    The information loss due to compression is defined as the conditional entropy of the original features given the compressed feature, i.e., $L_c = H(\{\mathbf{Y}_t\} \mid \mathbf{Z})$. By the **definition of conditional entropy**, for any random variables $A$ and $B$:

$$H(A \mid B) = H(A, B) - H(B), \tag{6}$$

where $H(A, B)$ is the joint entropy of $A$ and $B$, and $H(B)$ is the marginal entropy of $B$.

For our setting, let $A = \{\mathbf{Y}_1, \ldots, \mathbf{Y}_T\}$ (original features) and $B = \mathbf{Z}$ (compressed feature). The information loss thus becomes:

$$L_c = H(\{\mathbf{Y}_t\}, \mathbf{Z}) - H(\mathbf{Z}). \tag{7}$$

**Step 2: Simplifying $H(\{\mathbf{Y}_t\}, \mathbf{Z})$ for Deterministic Compression**    The compression operation $\mathcal{C}$ is **deterministic**, meaning $\mathbf{Z}$ is uniquely determined by $\{\mathbf{Y}_t\}$ (i.e., $\mathbf{Z} = \mathcal{C}(\{\mathbf{Y}_t\})$). For deterministic functions, the conditional entropy of the output given the input is zero:

$$H(\mathbf{Z} \mid \{\mathbf{Y}_t\}) = 0, \tag{8}$$

since no uncertainty remains about $\mathbf{Z}$ once $\{\mathbf{Y}_t\}$ is known.

By the chain rule of joint entropy:

$$H(\{\mathbf{Y}_t\}, \mathbf{Z}) = H(\{\mathbf{Y}_t\}) + H(\mathbf{Z} \mid \{\mathbf{Y}_t\}). \tag{9}$$

Substituting $H(\mathbf{Z} \mid \{\mathbf{Y}_t\}) = 0$, we get:

$$H(\{\mathbf{Y}_t\}, \mathbf{Z}) = H(\{\mathbf{Y}_t\}). \tag{10}$$

Combining eq. (7) and eq. (10), the information loss simplifies to:

$$L_c = H(\{\mathbf{Y}_t\}) - H(\mathbf{Z}). \tag{11}$$

**Step 3: Image Encoder Loss Derivation**    Image encoders assume **inter-frame independence**, i.e., the joint probability factorizes as:

$$p(\mathbf{Y}_1^{\text{img}}, \ldots, \mathbf{Y}_T^{\text{img}}) = \prod_{t=1}^{T} p(\mathbf{Y}_t^{\text{img}}). \tag{12}$$

For independent random variables, the **joint entropy equals the sum of marginal entropies**:

$$H(\mathbf{Y}_1^{\text{img}}, \ldots, \mathbf{Y}_T^{\text{img}}) = \sum_{t=1}^{T} H(\mathbf{Y}_t^{\text{img}}). \tag{13}$$

Substituting eq. (13) into eq. (11) (the general loss formula), the Image Encoder loss becomes:

$$L_c^{\text{img}} = \sum_{t=1}^{T} H(\mathbf{Y}_t^{\text{img}}) - H(\mathbf{Z}), \tag{14}$$

which matches eq. (3).

**Step 4: Video Encoder Loss Derivation**    Video encoders model **inter-frame dependencies** via spatiotemporal attention, so they do not assume independence. Their joint entropy is expanded using the **chain rule of joint entropy**:

$$H(\mathbf{Y}_1^{\text{vid}}, \ldots, \mathbf{Y}_T^{\text{vid}}) = \sum_{t=1}^{T} H(\mathbf{Y}_t^{\text{vid}} \mid \mathbf{Y}_1^{\text{vid}}, \ldots, \mathbf{Y}_{t-1}^{\text{vid}}), \tag{15}$$

where $H(\mathbf{Y}_t^{\text{vid}} \mid \mathbf{Y}_1^{\text{vid}}, \ldots, \mathbf{Y}_{t-1}^{\text{vid}})$ is the conditional entropy of the $t$-th frame given all previous frames.

By the **definition of mutual information**, for random variables $X$ and $\mathcal{Y}$:

$$I(X; \mathcal{Y}) = H(X) - H(X \mid \mathcal{Y}) \Longrightarrow H(X \mid \mathcal{Y}) = H(X) - I(X; \mathcal{Y}). \tag{16}$$

Applying this to the conditional entropy in eq. (15) (letting $X = \mathbf{Y}_t^{\text{vid}}$ and $\mathcal{Y} = \{\mathbf{Y}_1^{\text{vid}}, \ldots, \mathbf{Y}_{t-1}^{\text{vid}}\}$):

$$H(\mathbf{Y}_t^{\text{vid}} \mid \mathbf{Y}_1^{\text{vid}}, \ldots, \mathbf{Y}_{t-1}^{\text{vid}}) = H(\mathbf{Y}_t^{\text{vid}}) - I(\mathbf{Y}_t^{\text{vid}}; \mathbf{Y}_1^{\text{vid}}, \ldots, \mathbf{Y}_{t-1}^{\text{vid}}). \tag{17}$$

Substituting eq. (17) into eq. (15), the joint entropy for Video Encoders becomes:

$$H(\mathbf{Y}_1^{\text{vid}}, \ldots, \mathbf{Y}_T^{\text{vid}}) = \sum_{t=1}^{T} \left[ H(\mathbf{Y}_t^{\text{vid}}) - I(\mathbf{Y}_t^{\text{vid}}; \mathbf{Y}_1^{\text{vid}}, \ldots, \mathbf{Y}_{t-1}^{\text{vid}}) \right]. \tag{18}$$

Finally, substituting eq. (18) into eq. (11) (the general loss formula), the Video Encoder loss becomes:

$$L_c^{\text{vid}} = \sum_{t=1}^{T} \left[ H(\mathbf{Y}_t^{\text{vid}}) - I(\mathbf{Y}_t^{\text{vid}}; \mathbf{Y}_1^{\text{vid}}, \ldots, \mathbf{Y}_{t-1}^{\text{vid}}) \right] - H(\mathbf{Z}), \tag{19}$$

which matches eq. (4).

**Step 5: Comparing $L_c^{\text{img}}$ and $L_c^{\text{vid}}$**    For most videos, consecutive frames are correlated (e.g., static backgrounds or smooth motion), so the cumulative mutual information is positive:

$$I(\mathbf{Y}_t^{\text{vid}}; \mathbf{Y}_1^{\text{vid}}, \ldots, \mathbf{Y}_{t-1}^{\text{vid}}) > 0 \quad \forall t \geq 2. \tag{20}$$

This implies:

$$\sum_{t=1}^{T} H(\mathbf{Y}_t^{\text{vid}}) - \sum_{t=1}^{T} I(\mathbf{Y}_t^{\text{vid}}; \ldots) < \sum_{t=1}^{T} H(\mathbf{Y}_t^{\text{vid}}). \tag{21}$$

Given that Image and Video Encoders process the same video (so $\sum_{t=1}^{T} H(\mathbf{Y}_t^{\text{img}}) \approx \sum_{t=1}^{T} H(\mathbf{Y}_t^{\text{vid}})$) and share the same compressed feature $\mathbf{Z}$ (so $H(\mathbf{Z})$ is identical), we conclude:

$$L_c^{\text{img}} > L_c^{\text{vid}}. \tag{22}$$

Table 8: **Comparison of connectors.**

| Connector | MVBench Avg | MLVU M-Avg | VideoMME Overall | Avg |
|---|---|---|---|---|
| *#tokens per frame=729, compression ratio=100%* | | | | |
| MLP (Uncompressed) | 59.4 | 64 | 55.3 | 59.6 |
| *#tokens per frame=196, compression ratio=27%* | | | | |
| Spatial Downsampling | 60.2 | 63.7 | 52.8 | 58.9(-0.7) |
| Uneven Downsampling | 60.9 | 62.5 | 54.9 | 59.4(-0.2) |
| Spatio-temporal Resampler | 59.5 | 61.9 | 51.9 | 57.8(-1.8) |
| **Similar Token Merging** | **62.8** | **66.7** | **56.8** | **62.1**(+2.5) |
| *#tokens per frame=49, compression ratio=7%* | | | | |
| Spatial Downsampling | 60.2 | 61.8 | 53.6 | 58.5(-1.1) |
| Uneven Downsampling | 59.8 | 62.8 | 54.3 | 59.0(-0.6) |
| Spatio-temporal Resampler | 55.5 | 58.1 | 51.1 | 54.9(-4.7) |
| **Similar Token Merging** | **61.4** | **63.3** | **55.3** | **60.0**(+0.4) |
| *#tokens per frame=16, compression ratio=2%* | | | | |
| Spatial Downsampling | 58.1 | 61.1 | 50.1 | 56.4(-3.2) |
| Uneven Downsampling | 58.3 | 60.0 | 52.3 | 56.9(-2.7) |
| Spatio-temporal Resampler | 51.4 | 54.7 | 47.7 | 51.3(-8.3) |
| **Similar Token Merging** | **60.2** | **62.4** | **53.5** | **58.7**(-0.9) |

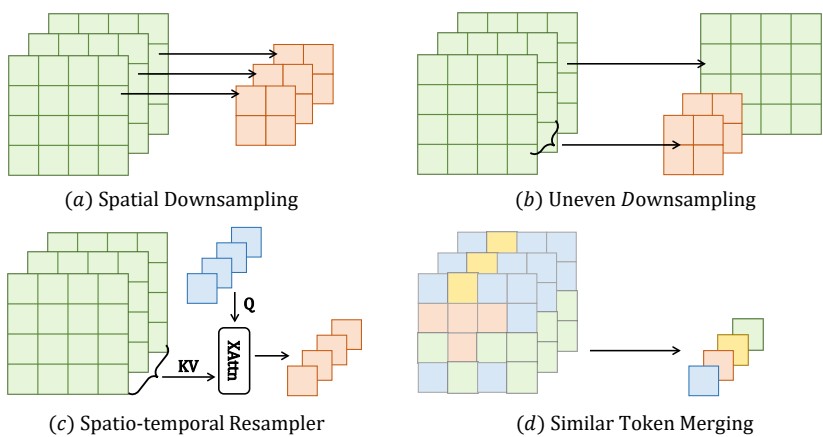

(a) Spatial Downsampling  (b) Uneven *Downsampling*

(c) Spatio-temporal Resampler  (d) Similar Token Merging

Figure 8: **Comparison of different connectors**.

## A.3 MORE RESULTS & DISCUSSIONS OF VIDEOCHAT-FLASH

### A.3.1 DETAILED RESULTS OF VISUAL COMPRESSION

**Different connectors and compression ratio.** As shown in the table 8, we consider three different numbers of tokens per frame after compression (16, 49, 196) and four popular token compression strategies: spatial downsampling (Zhang et al., 2024c; Chen et al., 2024e), uneven downsampling (Wei & Chen, 2024), spatio-temporal resampler (Li et al., 2024c; Tan et al., 2024), and similar token merging (Bolya et al., 2022; Weng et al., 2025) (more details can be found in the Appendix). It can be seen that compared with other methods, the parameter-free similar token merging operation can achieve a remarkably low compression ratio and even obtain better performance than without compression. Even in the extreme case of a 2% compression ratio, it can still maintain most of the performance.

Table 9: **Comparison of FLOPs and Cuda memory.** The FLOPs and inference memory is estimated using one NVIDIA A100-80G GPU with one sample, and the training is estimated using 32 NVIDIA A100-80G GPUs with DeepSpeed ZeRO-3 (Rasley et al., 2020). We assume that the visual features have been extracted and stored in advance, so we only consider the FLOPs and memory of the LLM.

| Input frames | Model | Avg tokens per frame | FLOPs (T) | Memory(G) Train | Infer |
|---|---|---|---|---|---|
| 64 | LongVILA (Xue et al., 2024) | 196 | 224.8 | 15.4 | 16.7 |
| | LongVA (Zhang et al., 2024a) | 144 | 155.9 | 12.3 | 16.3 |
| | **VideoChat-Flash** | **16** | **14.8** | **4.8** | **15.4** |
| 256 | LongVILA (Xue et al., 2024) | 196 | 1467.5 | 50.1 | 21.0 |
| | LongVA (Zhang et al., 2024a) | 144 | 930.4 | 37.8 | 19.5 |
| | **VideoChat-Flash** | **16** | **63.0** | **7.6** | **15.7** |
| 1000 | LongVILA (Xue et al., 2024) | 196 | 14336.9 | *oom* | 37.7 |
| | LongVA (Zhang et al., 2024a) | 144 | 8278.9 | *oom* | 31.8 |
| | **VideoChat-Flash** | **16** | **303.3** | **18.6** | **17.1** |
| 10000 | LongVILA (Xue et al., 2024) | 196 | 1184250.0 | *oom* | *oom* |
| | LongVA (Zhang et al., 2024a) | 144 | 644632.0 | *oom* | *oom* |
| | **VideoChat-Flash** | **16** | **9969.5** | *oom* | **33.6** |

**Training cost.** The main training advantage of our method lies in the fact that compression reduces the context sequence length fed into the LLM. First, we consider the impact of reduced context sequence length on training under the condition of the same training data volume. Here, we cite the comparison of training system throughput under different sequence lengths provided by the state-of-the-art long video training system LongVILA Xue et al. (2024) as a reference—this comparison is conducted on 64 H100 GPUs and measured in time per iteration (seconds).

Table 10: **Training system throughput comparison (cited from LongVILA).** The data is measured on 64 H100 GPUs, with the metric being time per iteration.

| Model | Token Per Frame | Max Frames | Sequence Length | Training Time per Iteration (s) |
|---|---|---|---|---|
| VideoChat-Flash | 16 | 512 | 8k | Not provided in LongVILA |
| | 16 | 2048 | 32k | 4.24 |
| InternVL2.5 | 256 | 512 | 128k | 16.0 |
| | 256 | 2048 | 512k | 66.1 |

As shown in table 10, it can be observed that despite the excellent acceleration optimizations for long-sequence training, changes in sequence length still significantly impact training speed. Our method can substantially reduce the training cost of long videos. Regarding the specific training cost, our method roughly requires 32 A100 GPUs for 5 to 6 days of training, which is significantly lower than that of mainstream models such as InternVL and QwenVL.

**Inference cost.** As in table 9, even when processing short videos, the compute load of our model is only one-tenth that of previous models. Meanwhile, as the number of input frames increases, the difference becomes more and more pronounced. Only our model can complete the inference on 10,000 frames on a single A100-80G. Concretely, VideoChat-Flash's compute load is two orders of magnitude lower than that of LongVILA (Xue et al., 2024) (9,969.5 vs. 1,184,250.0 TFLOPs). We also present the real inference costs using HuggingFace Pipeline and Flash-Attention2. As shown in table 11, under the same video frame input and resolution, the FLOPs, GPU memory consumption, and latency of our VideoChat-Flash are all significantly lower than those of Qwen2.5-VL—one of the representative mainstream MLLMs—demonstrating the high efficiency of our method.

### A.3.2 VISUAL DROPOUT IN LLM

**Visual token redundancy in LLM inference.** As shown in fig. 9, we find that even when half of the tokens are discarded at the shallow layers of the LLM, the performance of long video understanding

Table 11: **Inference cost comparison (measured via HuggingFace Pipeline with Flash-Attention2).**

| Model | Input Frames | Num. Visual Tokens | FLOPs | GPU Memory | latency (Vision Encoder) | latency (LLM) | latency (Total) |
|---|---|---|---|---|---|---|---|
| Qwen2.5-VL | 64 | 8192 | 158.65 | 18.66 | 1.115 | 2.308 | 3.422 |
| VideoChat-Flash | 64 | 1024 | 44.32 | 16.54 | 0.323 | 0.096 | 0.419 |
| Qwen2.5-VL | 256 | 32768 | 632.29 | 28.07 | 4.621 | 6.211 | 10.832 |
| VideoChat-Flash | 256 | 4096 | 175.03 | 20.41 | 1.188 | 0.337 | 1.526 |
| Qwen2.5-VL | 1024 | 131072 | 2526.80 | 65.72 | 18.930 | 34.473 | 53.403 |
| VideoChat-Flash | 1024 | 16384 | 697.81 | 35.92 | 4.697 | 1.442 | 6.140 |

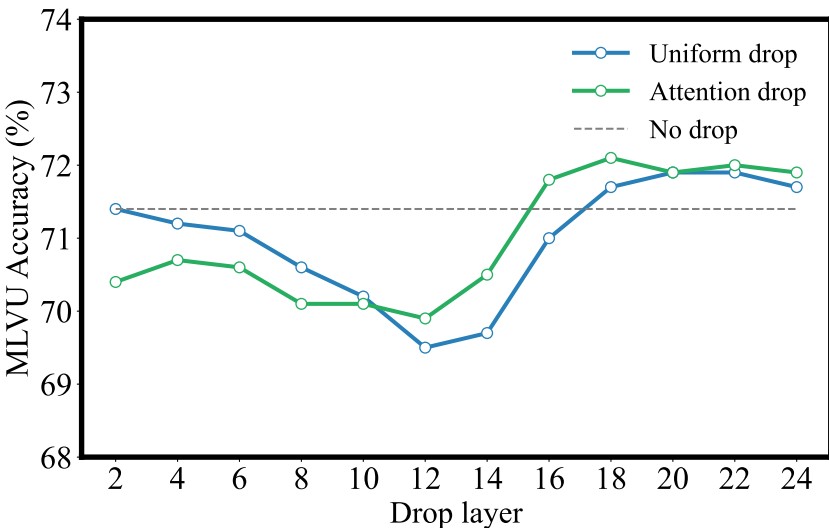

Figure 9: **Visual redundancy in long context across layers.** We conduct experiments on Qwen2-7B (28 layers) and test the impact of dropping 50% of the visual tokens from shallow to deep layers.

only degrades marginally. This indicates that despite high compression at the clip level (encoding each frame into only 16 tokens), there remains considerable redundancies between clips when their representations are interacted in the LLM. Furthermore, we find the overall understanding performance gets better as the dropout happens in the deeper layer of the model. Remarkably, at approximately two-thirds of the LLM's depth, the performance even surpasses that of the no-discard baseline. This might suggest that in the deeper layers of the network, an excess of visual tokens may interfere with the model's reasoning process. For the drop type, we observe that uniform drop often outperforms attention-based selection in the shallow layers. We suppose, at these layers, the LLM has not yet fully determined the specific locations to focus on. As a result, relying on attention may introduce bias.

**Visualization of visual attention map.** As shown in the fig. 10, for long video context, the attention of text tokens is relatively dispersed in the shallow layers of the network. However, as the layers deepen, the attention gradually becomes focused on specific regions. Thus, we believe that the attention scores in the deeper layers are more reliable, while those in the shallow layers may be prone to bias.

- **The Premise:** Effective dropping requires the model to accurately determine which visual tokens are semantically relevant to the text.

- **The Conflict:** At the early stages of training, the MLLM has not yet established robust alignment between textual and visual modalities. Consequently, it cannot reliably identify "key" visual information. Forcing the model to drop tokens based on unlearned or unstable attention weights prevents it from effectively learning the full video representation.

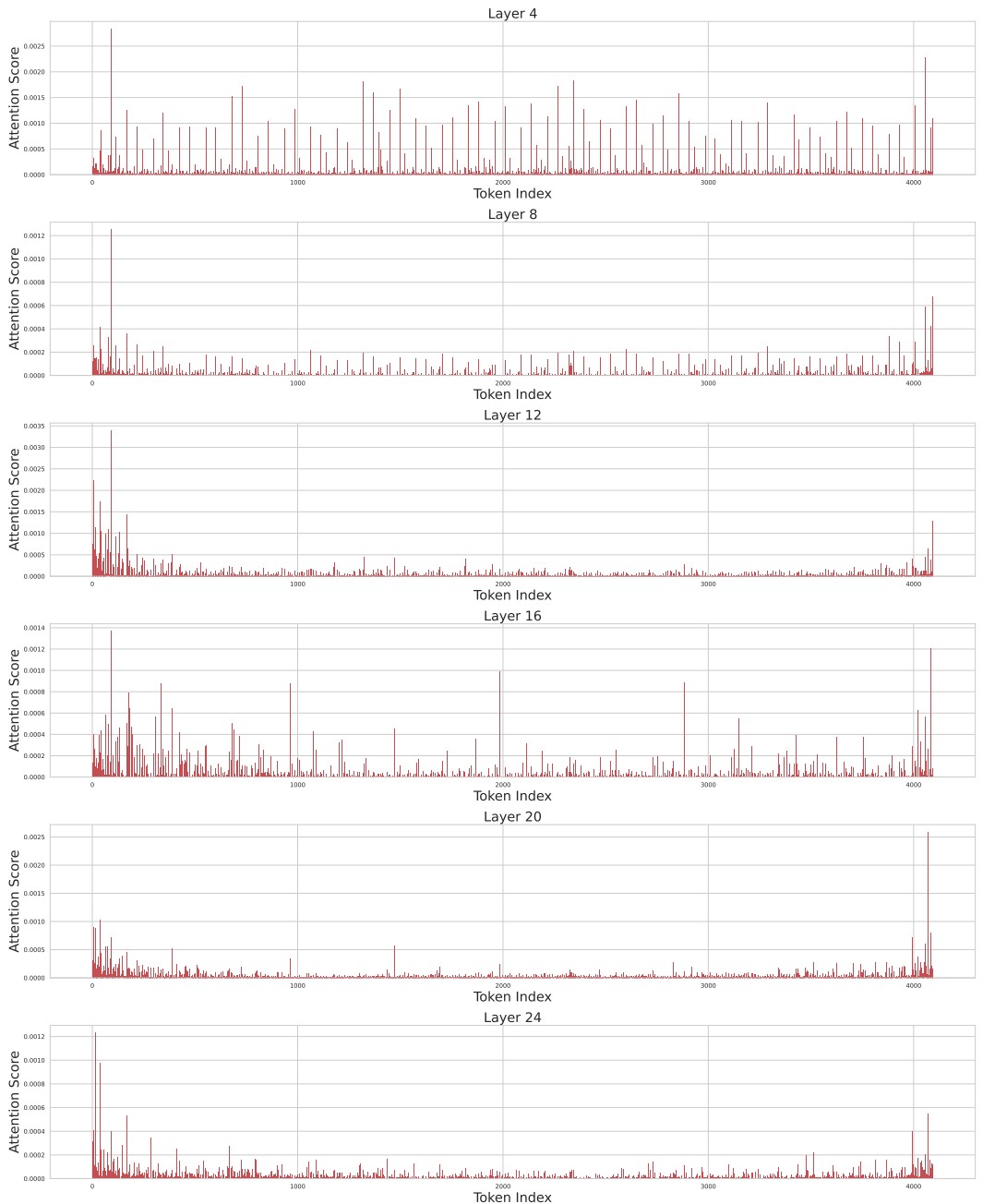

Figure 10: **Visualization of the attention scores from the last textual token to visual tokens at each layer of the network.**

### A.3.3 RESULTS ON IMAGE UNDERSTANDING BENCHMARKS

Our model is specifically designed for video understanding. However, according to the newly-evaluated results of image benchmarks, our model can still outperform the strong image-based MLLM, LLaVA-NeXT (Zhang et al., 2024c), with significantly lower computational cost: MMMU (45.2 vs. 35.3), MME (1843.4 vs 1603.7).

### A.3.4 THE IMPACT OF VISUAL COMPRESSION ON EFFICIENCY AND PERFORMANCE

we emphasize that the advantages of our compression strategy are twofold, benefiting both efficiency and performance:

- **From an efficiency perspective:** With the same number of input frames, our method significantly reduces computational costs without a marked decline in performance, as demonstrated in the second row of table 3.

- **From a performance perspective:** In scenarios where computational resources and LLM context length are constrained-a common bottleneck in long-video understanding where standard sparse sampling often leads to severe information loss-compression allows the model to ingest a much larger number of video frames. This enables the model to achieve better performance by scaling up the input video context. To further empirically validate this, we present the additional data below. As shown in the table 12, under a fixed budget of total Vision Tokens (4,096), compressing the per-frame representation (from 256 to 64 tokens) allows us to significantly increase the number of input frames (from 16 to 64). This enhanced temporal context leads to substantial performance gains.

Table 12: **Performance results under different Token per frame and Input Frames configurations.**

| Token Per Frame | Input Frames | Vision Token | VideoMME | MotionBench |
|---|---|---|---|---|
| 256 | 16 | 4096 | 51.3 | 44.8 |
| 64 | 64 | 4096 | 59.3(+8.0) | 46.2(+1.4) |

### A.3.5 FURTHER ELABORATION ON NOVELTY.

A critical distinction between our work and prior art lies in our objective: **we pursue efficient long-video modeling to build long-video MLLMs from scratch**, whereas most existing works aim to accelerate existing MLLMs (e.g., by freezing most modules or adopting training-free approaches).

We believe that learning native compressed representations for long-video modeling is essential. To this end, we propose the Hierarchical Compression (HiCo) paradigm. This is an **effective system-level design rather than merely a specific component**. HiCo decouples compression into two stages: handling visual redundancy at the Clip-level (within the ViT) and textual semantics at the Video-level (within the LLM). Furthermore, regarding specific design details, our approach offers key differences and novel insights compared to previous work:

- **Clip-level Compression:** We propose leveraging spatio-temporal attention to capture local inter-frame redundancy. This allows the Video Encoder to learn how to compress representations via parameter updates, rather than relying on hand-crafted compression modules. This self-adaptive mechanism is key to achieving our exceptionally high compression rate.

- **Video-level Compression:** We introduce a new improved visual drop scheme specifically optimized for the long-video understanding domain.

- **Timestamp Prompts:** We demonstrate that concise text prompts remain highly effective even under extreme visual compression. This avoids the computational overhead associated with adding text timestamps to every frame, a common practice in prior work.

- **Duration-based Sampling:** We validate a more effective sampling strategy: applying dense sampling for short videos and sparse sampling for long videos. This differs from

previous works, which predominantly utilize fixed FPS sampling or rely exclusively on sparse sampling.

In summary, by combining these improvements, we are the first to demonstrate that **heavy-compression-based methods can achieve long-video understanding performance comparable to, or even surpassing, closed-source context-extension-based models.**

### A.3.6 DISCUSSION ON THE TRAINING-FREE FRAME SELECTION METHOD

Numerous training-free frame selection approaches Tang et al. (2025); Zhu et al. (2025); Sun et al. (2025) have been employed to enhance the long-video understanding of MLLMs. These methods target the selection of key frames or clips most relevant to a specific task, thereby avoiding the direct input of the full long-video context. This improves efficiency and often boosts performance, particularly given that most MLLMs are currently optimized for short-context rather than long-context tasks.

While we acknowledge that training-free frame selection is an effective, low-cost solution for specific application scenarios, we argue that it possesses inherent limitations when viewed from the perspective of constructing a general-purpose foundation model for long video understanding:

- *Task Formulation & Intrinsic Capability:* Fundamentally, frame selection simplifies a long-video task into a short-context problem; it does not enhance the MLLM's intrinsic ability to process long temporal contexts. in contrast, our approach reduces the visual token count via clip-level compression and employs a short-to-long training strategy. This allows the MLLM to efficiently and holistically comprehend the compressed long-video context.

- *Granularity & Precision:* External selection modules typically have a limited performance ceiling. A text-guided selector may struggle to accurately interpret complex instructions. Conversely, our method utilizes video-level compression to leverage the LLM's inherent capabilities in identifying critical visual information. This achieves precise token-level selection, which offers significantly finer granularity than coarse frame-level selection.

- *Information Retention:* Frame selection inevitably requires discarding a significant portion of the raw data, making it unsuitable for tasks requiring dense, global understanding (e.g., dense video captioning or detailed video description). Our method retains essential semantic information, demonstrating superior performance on these information-intensive tasks.

In summary, we regard frame selection and our proposed method as distinct paradigms tailored for different objectives: the former is suitable for adapting MLLMs in a training-free manner, while the latter is designed to train a native long-video MLLM.

### A.4 IMPLEMENTATION DETAILS OF VIDEOCHAT-FLASH

### A.4.1 VIDEO-LANGUAGE CONNECTORS

As shown in fig. 8, we consider four popular token compression strategies to compress the features from video clips:

- *Spatial Downsampling.* Applying spatial operations (pooling (Xu et al., 2024), interpolation (Zhang et al., 2024c), and convolution (pixel shuffle) (Chen et al., 2024e)) to each video frame for downsampling has been demonstrated in previous work (Xu et al., 2024; Maaz et al., 2024) as an effective method to reduce the number of video tokens. However, due to the lack of temporal interaction, this approach fails to leverage the relation between frames. We use pixel shuffle in our experiments.

- *Uneven Downsampling.* Considering the similarities between adjacent frames, it is unnecessary to retain full details for every frame. We can apply down-sampling operations with different sizes across frames within a clip. Specifically, a lower down-sampling size is applied to the first frame, while higher down-sampling sizes are used for the remaining frames. Similar approaches have been validated in a recent study (Wei & Chen, 2024).

- *Spatio-Temporal Resampler.* Using a learnable compressor, such as a Q-Former (Li et al., 2024c) or a cross-attention layer, to compress spatiotemporal tokens. However, this approach

requires a large amount of data for effective learning. In training, we observe that the Q-Former barely converges well in our setting. So in our ablations, we adopt a single-layer cross-attention instead.

- ***Similar Token Merging.*** We directly merge similar tokens, using the ToMe (Bolya et al., 2022) approach.

### A.4.2 VISUAL DROPOUT IN LLM

Herein, we present the detailed implementation of visual dropout in LLM: For uniform drop, we uniformly drop a proportion of visual tokens and reassign position ids to the retained visual tokens. For text-guided selection, since Flash-Attention2 Dao (2023) does not support returning valid attention maps, we instead compute the attention scores between text tokens and vision tokens independently for text-guided selection.

### A.4.3 TRAINING HYPERPARAMETERS.

As shown in Table 1, the training details and hyperparameters for each stage of our VideoChat-Flash model are presented.

Table 13: **Training details of each training stage for the VideoChat-Flash-7B model.**

| | | Stage-1 | Stage-2 | Stage-3 | Stage-4 |
|---|---|---|---|---|---|
| *Vision* | **Resolution×Num. frames** | 224 | 224 ×8 | 224×(64∼512) | 224×(64∼512) |
| | #Tokens | 16×4 | 16×8 | 16×(64∼512) | 16×(64∼512) |
| *Data* | **Dataset** | Image & Short Video | Image & Short Video | (Multi)-Image & Short/Long Video | (Multi)-Image & Short/Long Video |
| | #Samples | 1M | 4M | 3.2M | 0.3M |
| *Model* | **Trainable** | Projector | Full Model | Full Model | ViT&Projector |
| | 7.6B LLM | 20.0M | 7.9B | 7.9B | 0.3B |
| *Training* | **Batch Size** | 512 | 256 | 256 | 256 |
| | **LR** of *vision encoder* | $1\times10^{-3}$ | $2\times10^{-6}$ | $2\times10^{-6}$ | $2\times10^{-6}$ |
| | **LR** of *connector & LLM* | $1\times10^{-3}$ | $1\times10^{-5}$ | $1\times10^{-5}$ | $1\times10^{-5}$ |
| | **Epoch** | 1 | 1 | 1 | 1 |

### A.4.4 TRAINING DATA

**Stage 1: Video-Language Alignment.** In this stage, we use 558k image-text pairs from LCS-558K (Liu et al., 2023) and 481k short video-text pairs from S-MiT (Monfort et al., 2021).

**Stage 2: Short Video Pre-training.** To enhance the model's understanding of visual concepts, we conduct visual pre-training using 3.5 million images and 2.5 million short video-text pairs.

- ***Video Description Data.*** We utilize the video description data recaptioned with VideoChat2 (Li et al., 2024c) from WebVid2M (Bain et al., 2021).

- ***Detailed Video Description Data***. We employ the 323k detailed video description data recaptioned with Gemini (Reid et al., 2024) from WebVid (Bain et al., 2021) and Kinetics (Kay et al., 2017), as in previous work (Share, 2024).

- ***Detailed Image Description Data.*** We use the 3.5 million detailed image description data recaptioned with LLava-NeXT-34B (Zhang et al., 2024c) from the following datasets: COCO118K, BLIP558K, and CC3M, as provided by previous work (Li et al., 2024a).

- ***Text Data.*** To enhance the model's language understanding capabilities, we incorporate 143K samples from the Evo-Instruct dataset (Chen et al., 2024a).

**Stage 3: Joint Short & Long Video Instruction tuning.** To enable the model to handle a wide variety of video tasks, we collect 3.5 million instruction fine-tuning samples, including 1.1M images, 1.7M short videos (under 60 seconds), and 0.7M long videos (60∼3600 seconds).

- ***Long Video Instruction data.*** We primarily utilized long video instruction data from MoiveChat (Song et al., 2024), Vript (Yang et al., 2024) and a subset of LongVid.

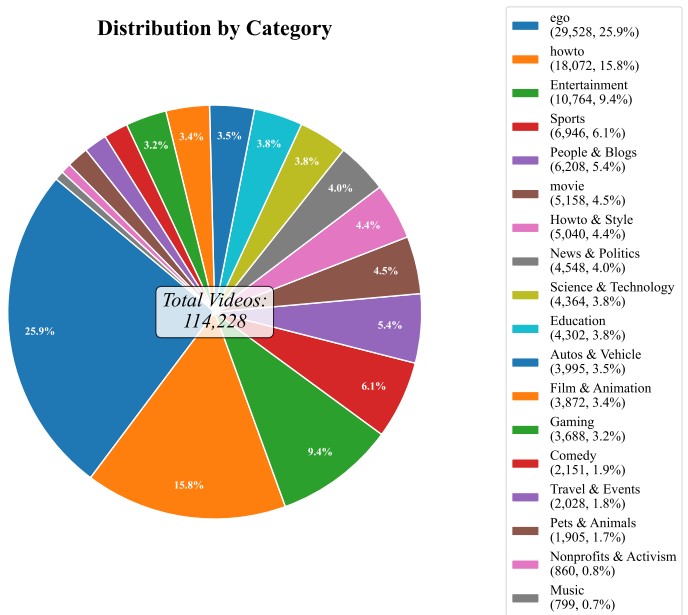

Figure 11: **Distribution by categroy.**

- ***Short Video Instruction data.*** We primarily utilized short video data from VideoChat2 (Li et al., 2024c) and InternVideo2 (Wang et al., 2024e) for instruction fine-tuning. Additionally, we incorporated data annotated with GPT4-o from previous works, including ShareGPT4o (Chen et al., 2024e; Wang et al., 2024e), VideoChatGPT-Plus (Maaz et al., 2024), LLaVA-Video-178K (Zhang et al., 2024d) and LLava-Hound (Zhang et al., 2024b).

- ***Image Instruction data.*** We primarily utilized single-image instruction data from LLava-NeXT (Zhang et al., 2024c), Allava (Chen et al., 2024a), and ShareGPT4-o (Chen et al., 2024e; Wang et al., 2024e). Additionally, we incorporated multi-image data provided by LLaVA-Interleave (Li et al., 2024b).

## A.5 DATASET DETAILS OF LONGVID

The videos of LongVid are curated from 4 open-source video datasets: Ego4D (Grauman et al., 2022), HowTo100M (Miech et al., 2019), HD-VILA (Xue et al., 2022), and MiraData (Ju et al., 2024). We provide statistics and details of the data construction pipeline for each dataset as follows.

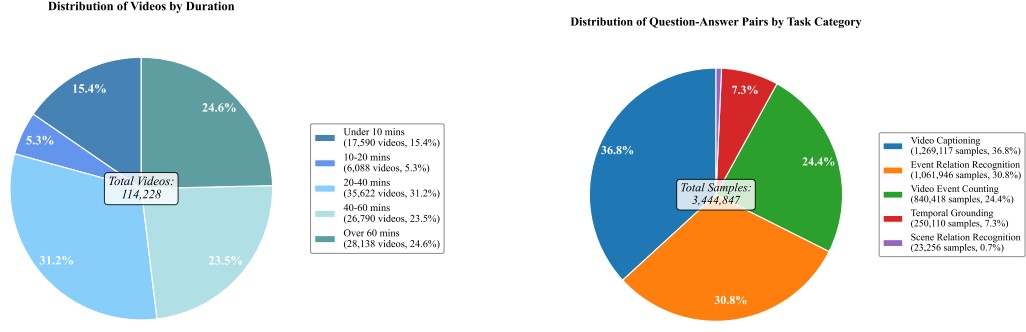

Figure 12: **Distribution of videos by duration.**

Figure 13: **Distribution of Question-Answer Pairs by Task Category.**

A.5.1    STATISTICS AND EXAMPLES

As shown in fig. 11, fig. 12 and fig. 13, we've shown the data and QA type distribution for LongVid. Next, we will provide examples for the five tasks.

**Task 1:**  Video Captioning

- **Question:** Which option best describe artistic style, visual and photographic aspects for this video, such as realistic, cyberpunk, and cinematic style?
- **Options&Answer:**
  (A)  The video features a high-definition, realistic graphic style with detailed textures and vibrant lighting effects, emphasizing a futuristic and immersive cyberpunk aesthetic.
  (B)  The video features a high-definition, realistic graphic style with detailed textures and dynamic lighting, emphasizing a gritty, futuristic aesthetic that is both immersive and visually engaging.
  (C)  **The video features a high-definition, realistic artistic style with a focus on detailed textures and vibrant lighting, enhancing the immersive medieval game setting.**
  (D)  The video features a realistic yet distinctly stylized graphic design typical of modern video games, with vibrant colors and detailed environments that enhance the immersive experience.

**Task 2:**  Temporal Grounding

- **Question:** When does a person is walking down a street in a video game?
- **Options&Answer:**
  (A)  **00:08:56.833 - 00:09:56.367**
  (B)  00:03:03.033 - 00:04:09.367
  (C)  00:07:21.900 - 00:08:07.633

**Task 3:**  Event Relation Recognition

- **Question:** What is the event in the video between "fold the wrapper loosely around the filling" and "repeat until there are three or four folds on each side"?
- **Options&Answer:**
  (A)  wrap the dough and let it sit 10–30 minutes
  (B)  mix the vegetables and ground meat together
  (C)  add the potstickers to the hot oil
  (D)  **fold a piece of dough at one corner**
  (E)  peel and chop other herbs and vegetables

**Task 4:**  Scene Relation Recognition

- **Question:** Which option can best describe the scenery of the video?
- **Options&Answer:**
  (A)  **lush → forest → sun → sunset**
  (B)  rural → suburban
  (C)  city night → streetlights → car → buildings → trees
  (D)  indoors → makeshift → operations room

**Task 5:**  Video Event Counting

- **Question:** According to the video, how many steps does the chef take during cooking?
- **Answer:**
  (A)  **6**

### A.5.2  EGO4D

For ego-centric videos, we adopt 3,662 long videos from the Ego4d (Grauman et al., 2022) and leverage Ego4DHcap (Islam et al., 2024) as the corresponding captions. Ego4DHcap gives hierarchical captions for short, medium, and long video segments. For the short video captioning task, we directly utilize these captions, while for the dense caption task, we concatenate captions in the lower level to form a dense one. For example, we merge all short video captions in a medium video segment to create a dense medium-level one, and the dense caption of long video segments can be formed by concatenating multiple medium-level video captions.

We also build event relation recognition and temporal grounding tasks based on captions of short video segments. For the event relation recognition task, models are required to choose the right order of an event sequence. Since we find the captions of short videos are highly concise and event-oriented, we use them as the event labels and serially put the short captions in a medium-level video segment as the ground-truth event relationship. For the temporal grounding task, we use the short video captions with the corresponding timestamps as the ground-truth, and randomly select other timestamps in the current medium video segments as the false options.

### A.5.3  MIRADATA

MiraData (Ju et al., 2024) provides multi-level captions for large-scale minute-level movie segments. Apart from short and dense captions that are used for short and dense video captioning tasks, it also provides multiple fine-grained captions that focus on various specific perspectives, such as the main object, background, camera movements, and video style. We use an open-source LLM (Qwen-72b (Bai et al., 2023)) to extract the event and background labels from the main object and background captions, respectively, and we put the labels of a long video in the right order as the ground truth of the event/background relation recognition task. For the temporal grounding task, we use the event label with the corresponding timestamp as the ground-truth option.

### A.5.4  HOWTO100M

HowTo100M (Miech et al., 2019) includes more than 1 million long-duration how-to videos. We adopt HowToInterlink7M (Wang et al., 2024a), a video captioning dataset that provides refined interleaved video captions of HowTo100M videos as short and dense video captions. For the event relationship recognition and temporal grounding tasks, we use HTStep (Afouras et al., 2024), a large-scale dataset containing temporal annotations of instructional steps in HowTo100M videos.

### A.5.5  HD-VILA

While previous datasets focus on long videos in specific domains, we also select part of the videos from HD-VILA (Xue et al., 2022), a large-scale video dataset that includes various in-the-wild videos. We argue that adding these videos into training could enhance the model's ability to process long videos in some uncommon domains. For HD-VILA videos, we adopt the captions of Panda-70M (Chen et al., 2024c). Specifically, we filter consecutive video segments that can be re-constructed into more than 60s long videos from the 10M training subset and utilize these captions as the video short/dense captioning and temporal grounding tasks. The event labels are also extracted from these captions in the same way as MiraData (Ju et al., 2024).

### A.6  QUALITATIVE RESULTS OF VIDEOCHAT-FLASH

We perform qualitative comparisons of our model with the proprietary model Gemini-1.5 Pro (Reid et al., 2024)[1] and the open-source LongVU (Shen et al., 2024) and VideoLLaMA2 (Cheng et al., 2024) across three tasks: fine-grained understanding of short videos ( figs. 14 and 15) and long video understanding ( figs. 16 and 17).

---

[1] We use the newest Gemini-1.5 Pro-002 for evaluation.

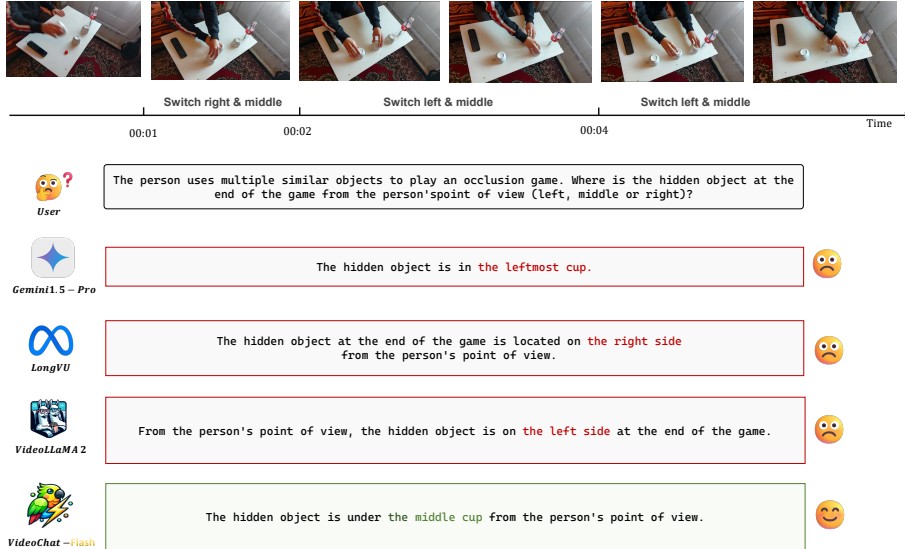

Figure 14: **Fine-grained Understanding of Short Videos: Fast Motion.** By adopting a dense sampling strategy for short videos, our model effectively captures fast motion within the video, enabling it to accurately determine the final position of the object under the cup.

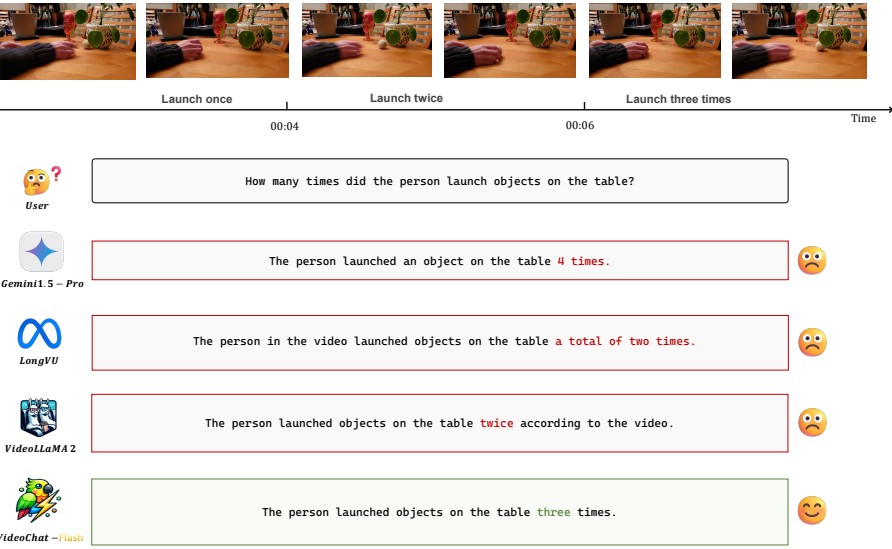

Figure 15: **Fine-grained Understanding of Short Videos: Action Count.** Our model can accurately capture actions in short videos while also recording their occurrence frequency.

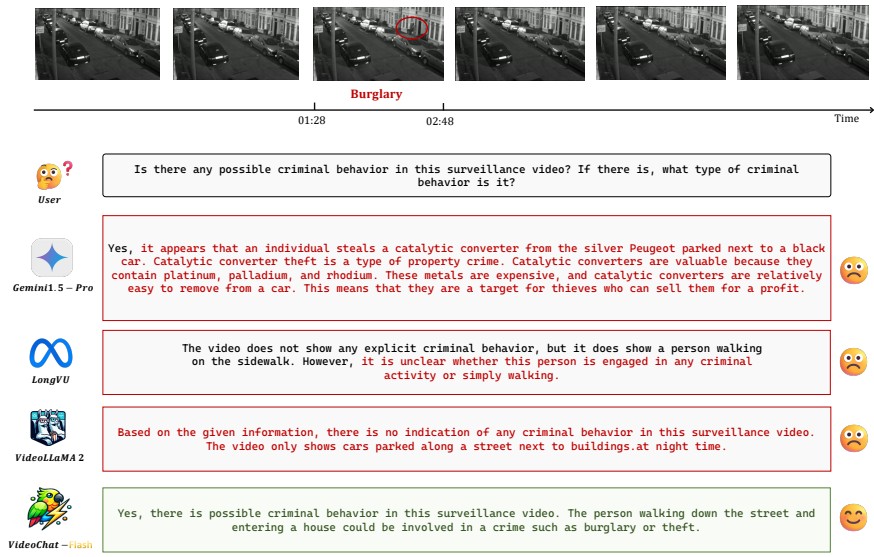

Figure 16: **Long video understanding: Anomaly Detection In Surveillance Videos.** Our model can detect anomalous behaviors in surveillance videos and provide corresponding inferences.

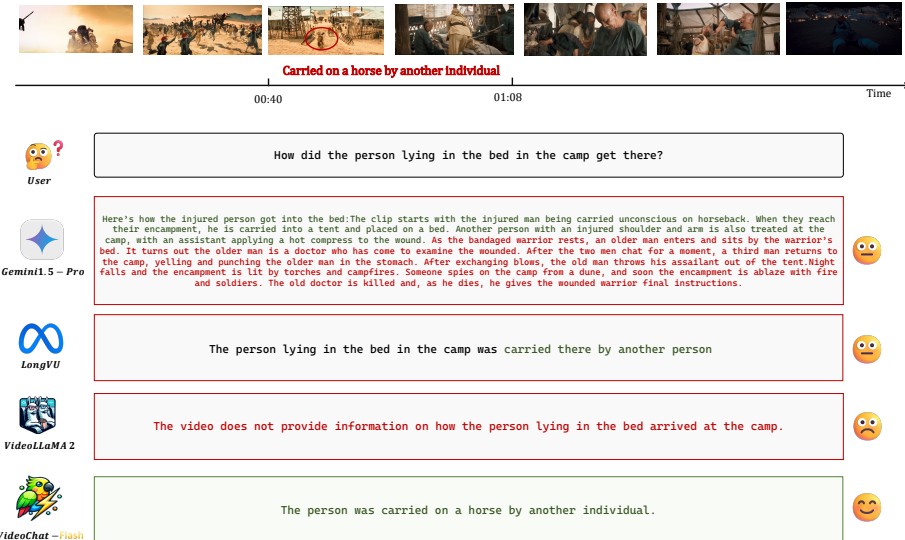

Figure 17: **Long video understanding: Moive Understanding.** Our model can understand the plot of a movie and retain detailed visuals.

