# OpenReview forum: "VideoChat-Flash: Hierarchical Compression for Long-Context Video Modeling"
_ICLR.cc/2026/Conference — ICLR 2026 Poster_

### Official Review · Reviewer_m8kd · 2025-10-28

**Soundness:** 3
**Presentation:** 3
**Contribution:** 3
**Rating:** 8
**Confidence:** 3

**Summary:**

This paper introduces VideoChat-Flash, a new model for long-context video modeling. It is trained on LongVid, a novel dataset combining Ego4D, HowTo100M, HD-Vila, and MiraData, with additional dense and timestamped event labels. The core of their method is hiCo (Hierarchical Compression), a multi-step technique to reduce both clip-level and video-level redundancy. It involves sampling clips based on video duration, using a vision encoder with spatio-temporal attention to compress visual and temporal information within clips, and then merging these compressed features into a final vision context aligned with LLM representations via a projection head. The training process is multi-staged: video-language alignment, pre-training with short videos, short and long video instruction tuning, and finally, post-finetuning for higher-resolution video encoding. The authors also present a new "Needle in a Haystack" benchmark. Unlike traditional benchmarks that insert a single needle, their approach leverages model reasoning by embedding multiple clues in a video that the model must follow to find the final objective. VideoChat-Flash significantly outperforms models like GPT4, Gemini 1.5, and other open-source models on various benchmarks, including LongVideoBench, MLVU, VideoMME, and LVBench, demonstrating strong performance across long video contexts. An ablation study was conducted to analyze different compression ratios and architectural and training criterion designs.

**Strengths:**

- Paper well-written
- Lot of artefacts such as new model, new training data mix, and new evaluation protocole
- Really like the idea of a reasoning based needle in a video haystack evaluation
- Extensive experimental protocole and comparison against lot of different methods
- Performance significantly higher than the concurrent methods
- In depth ablation studies over the compression effect, sampling strategies, video encoder and design choices
- Good appendix with details on hyper-parameters that were used, additional ablations studies and additional training data mix information.

**Weaknesses:**

- Would have expected some discussions over the overall training/inference cost of this method in contrast to other.
- Missing some discussion on the training-free frame selection method such as AKS (Adaptive Keyframe Sampling for Long Video Understanding)
- Missing information about the different video data licenses. A datasheet would have been appreciated.

**Questions:**

- Do the authors plan to open-source models and benchmarks?
- For the Multi-Hop Needle-In-A-Video-Haystack eval, are the clues always going forward in the video or can they also be backward? Let's say you have 3 hops, one at 5s, another at 8s, and the last one at 3s before having all the clues to get to the final haystack.

---

> ### Author Response · Authors · 2025-11-22
>
> ## Q1. Would have expected some discussions over the overall training/inference cost of this method in contrast to other.
>
> Regarding training cost, the main training advantage of our method lies in the fact that compression reduces the context sequence length fed into the LLM. First, we consider the impact of reduced context sequence length on training under the condition of the same training data volume. Here, we cite the comparison of training system throughput under different sequence lengths provided by the state-of-the-art long video training system LongVILA [1] as a reference—this comparison is conducted on 64 H100 GPUs and measured in time per iteration (seconds):
>
> | Model           | Token Per Frame | Max Frames | Sequeue Length | Training time per iteration (s) |
> | --------------- | --------------- | ---------- | -------------- | ------------------------------- |
> | VideoChat-Flash | 16              | 512        | 8k             | Not provided in LongVILA        |
> |                 | 16              | 2048       | 32k            | 4.24                            |
> | InternVL2.5     | 256             | 512        | 128k           | 16.0                            |
> |                 | 256             | 2048       | 512k           | 66.1                            |
>
> It can be observed that despite the excellent acceleration optimizations for long-sequence training, changes in sequence length still significantly impact training speed. Our method can substantially reduce the training cost of long videos. Regarding the specific training cost, our method roughly requires 32 A100 GPUs for 5 to 6 days of training, which is significantly lower than that of mainstream models such as InternVL and QwenVL.
>
> **Regarding inference cost**, we present the inference costs using HuggingFace Pipeline and Flash-Attention2:
>
> | **Model**       | **Input Frames** | **Num. Visual Tokens** | **FLOPs** | **GPU** **memory** | latency（Vision Encoder) | **latency（LLM）** | **latency（Total）** |
> | --------------- | ---------------- | ---------------------- | --------- | ------------------ | ------------------------ | ------------------ | -------------------- |
> | Qwen2.5-VL      | 64               | 8192                   | 158.65    | 18.66              | 1.115                    | 2.308              | 3.422                |
> | VideoChat-Flash | 64               | 1024                   | 44.32     | 16.54              | 0.323                    | 0.096              | 0.419                |
> | Qwen2.5-VL      | 256              | 32768                  | 632.29    | 28.07              | 4.621                    | 6.211              | 10.832               |
> | VideoChat-Flash | 256              | 4096                   | 175.03    | 20.41              | 1.188                    | 0.337              | 1.526                |
> | Qwen2.5-VL      | 1024             | 131072                 | 2526.80   | 65.72              | 18.930                   | 34.473             | 53.403               |
> | VideoChat-Flash | 1024             | 16384                  | 697.81    | 35.92              | 4.697                    | 1.442              | 6.140                |
>
> As observed under the same video frame input and resolution, the FLOPs, GPU memory consumption, and latency of our VideoChat-Flash are all significantly lower than those of Qwen2.5-VL—one of the representative mainstream MLLMs—demonstrating the high efficiency of our method.
>
> [1] LongVILA: Scaling Long-Context Visual Language Models for Long Videos

---

> > ### Author Response · Authors · 2025-11-22
> >
> > ## Q2. Missing some discussion on the training-free frame selection method such as AKS.
> >
> > We sincerely appreciate this insightful suggestion. Here, we provide a detailed elaboration on our rationale.
> >
> > As you correctly pointed out, numerous training-free frame selection approaches[1, 2, 3] have been employed to enhance the long-video understanding of MLLMs. These methods target the selection of key frames or clips most relevant to a specific task, thereby avoiding the direct input of the full long-video context. This improves efficiency and often boosts performance, particularly given that most MLLMs are currently optimized for short-context rather than long-context tasks.
> >
> > While we acknowledge that training-free frame selection is an effective, low-cost solution for specific application scenarios, we argue that it possesses inherent limitations when viewed from the perspective of constructing a **general-purpose foundation model** for long video understanding:
> >
> > - **Task Formulation & Intrinsic Capability:** Fundamentally, frame selection simplifies a long-video task into a short-context problem; it does not enhance the MLLM’s intrinsic ability to process long temporal contexts. in contrast, our approach reduces the visual token count via clip-level compression and employs a short-to-long training strategy. This allows the MLLM to efficiently and holistically comprehend the compressed long-video context.
> > - **Granularity & Precision:** External selection modules typically have a limited performance ceiling. A text-guided selector may struggle to accurately interpret complex instructions. Conversely, our method utilizes video-level compression to leverage the LLM's inherent capabilities in identifying critical visual information. This achieves precise **token-level selection**, which offers significantly finer granularity than coarse frame-level selection.
> > - **Information Retention:** Frame selection inevitably requires discarding a significant portion of the raw data, making it unsuitable for tasks requiring dense, global understanding (e.g., dense video captioning or detailed video description). Our method retains essential semantic information, demonstrating superior performance on these information-intensive tasks.
> >
> > In summary, we regard frame selection and our proposed method as distinct paradigms tailored for different objectives: the former is suitable for adapting MLLMs in a training-free manner, while the latter is designed to train a native long-video MLLM.
> >
> >
> >
> > [1] Adaptive Keyframe Sampling for Long Video Understanding
> >
> > [2] FOCUS: Efficient Keyframe Selection for Long Video Understanding
> >
> > [3] From Frames to Clips: Efficient Key Clip Selection for Long-Form Video Understanding
> >
> > ## Q3. Missing information about the different video data licenses. A datasheet would have been appreciated.
> >
> > We appreciate your suggestions regarding the dataset licenses.  The dataset utilized in our research is collected from publicly accessible sources, all of which are licensed under Creative Commons (CC-BY) or other open-source licenses.  We have diligently adhered to all legal requirements for integrating this data into our work, emphasizing the significance of transparency in data licensing to ensure proper attribution and appropriate usage.  We are committed to maintaining a high-quality and ethically sound dataset, and we pledge to uphold principles of privacy and transparency throughout all stages of our research.  Additionally, upon acceptance of the paper, we will supplement and open-source all details related to the training data, including the corresponding datasheet.
> >
> > ## Q4. Do the authors plan to open-source models and benchmarks?
> >
> > We thank the reviewer for this important question regarding reproducibility. Yes, we are fully committed to open-sourcing all models, code, training data, and benchmarks used in this work, as we hope to make a valuable contribution to the research and open-source communities.
> >
> > ## Q5. For the Multi-Hop Needle-In-A-Video-Haystack eval, are the clues always going forward in the video or can they also be backward? Let's say you have 3 hops, one at 5s, another at 8s, and the last one at 3s before having all the clues to get to the final haystack.
> >
> > We randomly shuffled the order of clues for each question. That is to say, the model cannot answer the questions through simple forward or backward sequential scanning; instead, it needs to continuously jump across different temporal positions in the video to search for clues. We supplemented the results of Gemini 2.5 Flash and found that the thinking mechanism is highly effective for our task, which also indicates to a certain extent that our task is complex and requires repeated thinking to complete.
> >
> > | Model   | Thinking | **Cap** Score | **QA** Score |
> > | --------------- | -------- | ------------- | ------------ |
> > | Gemini2.5 Flash | ×        | 35%           | 31%          |
> > |                 | √        | 60%           | 54%          |

---

### Official Review · Reviewer_TNAS · 2025-10-30

**Soundness:** 3
**Presentation:** 3
**Contribution:** 3
**Rating:** 6
**Confidence:** 5

**Summary:**

This work aims to efficiently understand the extremely long video context with multimodal large language models (MLLMs). They propose a novel Hierarchical video token Compression (HiCo) method, which leverages visual redundancy in long videos to compress long video context from Clip-level to Video-level, reducing the computation significantly while preserving essential details. Then, they introduce a multi-stage short-to-long learning scheme and also a large-scale video dataset called LongVid as well as a benchmark. They build VideoChat-Flash that shows a leading performance on both mainstream long and short video benchmarks at the 2B and 7B model scales.

**Strengths:**

1. The topic is meaningful to the community: we need to find a better way to understand long videos with MLLMs efficiently.
2. The paper is well-organized and easy to follow. The experiments are solid and clear.
3. I really appreciate the authors for the LongVid dataset and benchmark. Open-sourced data is important in this community.

**Weaknesses:**

1. The paper claims “almost no performance loss” under extreme 1/50 compression, yet the quantitative analysis (Fig. 6b) provides limited breakdown across different downstream tasks. It would be helpful if they can provide the results on temporal grounding or motion-sensitive tasks besides the three benchmarks.
2. The claimed efficiency benefits (1/50 token reduction) are shown in FLOP counts but not in end-to-end wall-clock latency or GPU memory consumption during real long video inference. It would be helpful to report them in the paper as well!
3. Although the LongVid dataset is described as large-scale and diverse, construction heavily relies on caption-based pseudo-labeling using other LLMs. There is limited discussion of data noise, filtering quality, or potential overlap with evaluation benchmarks.

**Questions:**

N/A.
One tiny suggestion: fig -> Fig, tab -> Tab

---

> ### Author Response · Authors · 2025-11-22
>
> ## Q1. It would be helpful if they can provide the results on temporal grounding or motion-sensitive tasks besides the three benchmarks.
>
> Since the model training data under the ablation settings in Figure 6 of the main text did not include the temporal grounding task, the model cannot perform the temporal grounding task effectively. To better observe the impact of the compression ratio on tasks such as temporal grounding, we supplemented the relevant data and reconducted the ablation experiments. Due to time and resource constraints, we were only able to complete two sets of experiments to investigate the influence of the compression ratio on temporal grounding and motion-sensitive tasks.
>
> | **Model**      | **Compression ratio** | **Charade-STA (m-IoU)** | **MotionBench** [1] |
> | -------------- | --------------------- | ----------------------- | ------------------- |
> | Qwen-2.5-VL-8B | 25%                   | 43.6                    | 53.0                |
> | Our Ablation   | 7%                    | 42.27                   | 58.6                |
> |                | 2%                    | 42.36                   | 56.7                |
>
> It can be observed that high compression ratios do not significantly affect temporal grounding and motion-sensitive tasks. Furthermore, we have supplemented the comparative results between our final model and state-of-the-art models, which corroborates from another perspective that our model still maintains excellent fine-grained temporal understanding capability while achieving high compression.
>
> | **Model**          | **Avg tokens per frame** | **Charade-STA (m-IoU)** | **MotionBench** [1] | **FAVOR-Bench** [2] |
> | ------------------ | ------------------------ | ----------------------- | ------------------- | ------------------- |
> | InternVL2.5        | 256                      | 9.82                    | 53.5                | 34.59               |
> | Qwen2.5-VL         | 1924                     | 43.6                    | 53.0                | 40.76               |
> | VideoChat-Flash-7B | 16                       | 48.0                    | 60.6                | 43.82               |
>
> [1] MotionBench: Benchmarking and Improving Fine-grained Video Motion Understanding for Vision Language Models
>
> [2] FAVOR-Bench: A Comprehensive Benchmark for Fine-Grained Video Motion Understanding
>
> ## Q2. The claimed efficiency benefits (1/50 token reduction) are shown in FLOP counts but not in end-to-end wall-clock latency or GPU memory consumption during real long video inference. It would be helpful to report them in the paper as well!
>
> We appreciate your suggestions. Herein, we present the inference costs using HuggingFace Pipeline and Flash-Attention2:
>
> | **Model**       | **Input Frames** | **Num. Visual Tokens** | **FLOPs** | **GPU** **memory** | latency（Vision Encoder) | **latency（LLM）** | **latency（Total）** |
> | --------------- | ---------------- | ---------------------- | --------- | ------------------ | ------------------------ | ---------------------- | ------------------------ |
> | Qwen2.5-VL      | 64               | 8192                   | 158.65    | 18.66              | 1.115                    | 2.308                  | 3.422                    |
> | VideoChat-Flash | 64               | 1024                   | 44.32     | 16.54              | 0.323                    | 0.096                  | 0.419                    |
> | Qwen2.5-VL      | 256              | 32768                  | 632.29    | 28.07              | 4.621                    | 6.211                  | 10.832                   |
> | VideoChat-Flash | 256              | 4096                   | 175.03    | 20.41              | 1.188                    | 0.337                  | 1.526                    |
> | Qwen2.5-VL      | 1024             | 131072                 | 2526.80   | 65.72              | 18.930                   | 34.473                 | 53.403                   |
> | VideoChat-Flash | 1024             | 16384                  | 697.81    | 35.92              | 4.697                    | 1.442                  | 6.140                    |
>
> As observed under the same video frame input and resolution, the FLOPs, GPU memory consumption, and latency of our VideoChat-Flash are all significantly lower than those of Qwen2.5-VL—one of the representative mainstream MLLMs—demonstrating the high efficiency of our method.

---

> > ### Author Response · Authors · 2025-11-22
> >
> > ## Q3. For LongVid, There is limited discussion of data noise, filtering quality, or potential overlap with evaluation benchmarks.
> >
> > Thank you for the suggestion. We have supplemented the paper with the following details:
> >
> > **Regarding Data Leakage:** We strictly enforced data decontamination. For YouTube-sourced benchmarks (e.g., VideoMME, LongVideoBench), we ensured no test set Video IDs appeared in our training data. For academic benchmarks (e.g., MVBench), we strictly avoided using official test splits. While we acknowledge that widely used sources (like Ego4D) may share similar distributions across splits, we addressed this by prioritizing out-of-domain benchmarks (e.g., VideoMME) in our ablation studies to ensure robust evaluation.
> >
> > **Regarding Data Noise:** We recognize that noise is difficult to eliminate entirely due to the high cost of manual long-video annotation. To mitigate this, we utilized a pipeline of LLMs and MLLMs for automated labeling and filtering. We believe that despite inherent noise, synthetic data is currently the most scalable solution for long-video understanding,  and we are committed to continuously iterating on this synthesis pipeline to further optimize data quality in future work.

---

### Official Review · Reviewer_cNh3 · 2025-10-31

**Soundness:** 3
**Presentation:** 3
**Contribution:** 4
**Rating:** 8
**Confidence:** 4

**Summary:**

The paper proposes a hierarchical video token compression method designed to efficiently handle long-video contexts without compromising performance. It begins with duration-based sampling, which adapts to the differing requirements of short and long videos. This is followed by clip-level compression to aggregate key information across frames, and finally, a progressive visual dropout mechanism that allocates attention differently across shallow and deep layers. The authors also introduce a training dataset aimed at enhancing video understanding capabilities along with a multi-stage training strategy. In addition, a new challenging evaluation benchmark is presented to better assess complex reasoning abilities. Finally, the model is evaluated on multiple benchmarks, demonstrating the effectiveness of the proposed method, with ablation studies further validating the contribution of each component.

**Strengths:**

This is a well-structured and thoroughly executed paper. It presents extensive and insightful experiments and analyses that may also benefit future work in video understanding. The proposed HiCo framework demonstrates strong and consistent performance across multiple benchmarks. Additionally, the authors provide a new training dataset and evaluation benchmark, which will further benefit the community and advance progress in video understanding. The comprehensive ablation studies clearly illustrate the contribution of each component.

**Weaknesses:**

1. Some recently released baseline video models [1][2][3][4] are not included in the comparison. It would improve readability if the authors could explicitly indicate which model represents the previous state of the art in Figure 1.
2. It would be beneficial to include more advanced video models as baselines for comparison on the multi-hop NIAH task. Presenting the performance of closed-source models, such as Gemini 2.5 Pro, could also provide valuable context for readers to comprehend the complexity of this task.
3. It would be helpful to provide more details about the text-guided selection mentioned in L214.

[1] Comanici, G., Bieber, E., Schaekermann, M., Pasupat, I., Sachdeva, N., Dhillon, I., Blistein, M., Ram, O., Zhang, D., Rosen, E. and Marris, L., 2025. Gemini 2.5: Pushing the frontier with advanced reasoning, multimodality, long context, and next generation agentic capabilities. arXiv preprint arXiv:2507.06261.

[2] Zhang, B., Li, K., Cheng, Z., Hu, Z., Yuan, Y., Chen, G., Leng, S., Jiang, Y., Zhang, H., Li, X. and Jin, P., 2025. Videollama 3: Frontier multimodal foundation models for image and video understanding. arXiv preprint arXiv:2501.13106.

[3] Wang, X., Si, Q., Wu, J., Zhu, S., Cao, L. and Nie, L., 2025. Adaretake: Adaptive redundancy reduction to perceive longer for video-language understanding. arXiv preprint arXiv:2503.12559.

[4] Xu, M., Gao, M., Li, S., Lu, J., Gan, Z., Lai, Z., Cao, M., Kang, K., Yang, Y. and Dehghan, A., 2025. Slowfast-llava-1.5: A family of token-efficient video large language models for long-form video understanding. arXiv preprint arXiv:2503.18943.

**Questions:**

1. Regarding the creation of the Multi-Hop NIAH evaluation task, is there any restriction on the maximum number of frames that can be inserted?
2. For the image encoder, what is the rationale for using SigLIP instead of SigLIP2, which is generally considered a stronger image encoder? Similarly, for the video encoder, why not choose VideoPrism?
3. Why does the accuracy drop when increasing $T_{min}$ from 64 to 128 in Figure 7 (a)? Intuitively, performance should improve as more frames are used to capture finer details, as suggested by the trend in Figure 7 (b).

---

> ### Author Response · Authors · 2025-11-22
>
> ## Q1. Some recently released baseline video models are not included in the comparison.
>
> Thank you for your valuable suggestions. The "Previous SoTA" presented in Figure 1 refers to the state-of-the-art (SoTA) models at the time of the completion of our work. We hereby supplement the performance comparisons of the latest models you mentioned, and will subsequently adjust the space accordingly to incorporate them into the main text.
>
> | Model                      | Base Model  | PerceptionTest | VideoMME | LongVideoBench | MLVU | LVBench |
> | -------------------------- | ----------- | -------------- | -------- | -------------- | ---- | ------- |
> | Gemini2.5 Pro              | -           | 77.3           | 82.0     | -              | -    | 68.2    |
> | Gemini2.5 Flash            | -           | 71.2           | 77.8     | -              | -    | 60.9    |
> | VideoLLama3                | Qwen2.5-7B  | 72.8           | 66.2     | 59.8           | 73.0 | 45.3    |
> | AdaReTake                  | Qwen2-VL-7B | -              | 64.2     | 57.2           | 72.0 | 48.9    |
> | Slowfast-llava-1.5         | Qwen2.5-7B  | 69.6           | 63.9     | 62.5           | 71.5 | 45.3    |
> | **VideoChat-Flash (Ours)** | Qwen2-7B    | 76.2           | 65.3     | 64.7           | 74.7 | 48.2    |
>
> As can be observed, given that our model was trained at an earlier stage and adopts a relatively older version of the language model (Qwen2) as its base model, its performance nevertheless maintains strong competitiveness with more recent models. This, to a certain extent, demonstrates the effectiveness of our proposed method, training strategy, and data construction.
>
> ## Q2. It would be beneficial to include more advanced video models as baselines for comparison on the multi-hop NIAH task.
>
> Thank you for your valuable suggestions. We have supplemented the results of Gemini 2.5 Flash and Gemini 2.5 Flash thinking on our MH-NIAH benchmark. Due to budget constraints, we only evaluated its performance under the 1000-frame setting, with an input token count of approximately 266k.
>
> | **Model**                  | Thinking | **Token Per Frame** | **Cap Score** | **QA Score** |
> | -------------------------- | -------- | ------------------- | ------------- | ------------ |
> | random                     | -        | -                   | 25%           | 6.25%        |
> | LlamaVid                   | ×        | 2                   | 20%           | 7%           |
> | LongVA                     | ×        | 144                 | 25%           | 18%          |
> | **VideoChat-Flash (Ours)** | ×        | 16                  | 33%           | 27%          |
> | Gemini2.5 Flash            | ×        | 258                 | 35%           | 31%          |
> |                            | √        | 258                 | 60%           | 54%          |
>
> Surprisingly, without enabling the reasoning mode, Gemini 2.5 Flash achieved a score only slightly higher than that of our VideoChat-Flash in the Multi-Hop NIAH test. However, its score significantly improved when the thinking mode was activated. This further validates the fact that **the Multi-Hop NIAH task we designed truly requires more complex video reasoning capabilities rather than mere retrieval abilities for successful completion**.
>
> ## Q3. It would be helpful to provide more details about the text-guided selection mentioned in L214.
>
> We appreciate your suggestions. In Appendix A.4 of the latest version, we have supplemented additional detailed descriptions. Furthermore, we will incorporate more details into the future final version.
>
> ## Q4. Regarding the creation of the Multi-Hop NIAH evaluation task, is there any restriction on the maximum number of frames that can be inserted?
>
> For each question, we incorporated 5–8 images (including "real needles" and distractors). We found that this setup is sufficiently challenging. moreover, given the low cost of our construction method, we can further enhance the complexity of the tasks by increasing the number of incorporated images.
>
> ## Q5. For the image encoder, what is the rationale for using SigLIP instead of SigLIP2, which is generally considered a stronger image encoder? Similarly, for the video encoder, why not choose VideoPrism?
>
> At the time of our experiments, SigLIP2 had not yet been released, and similarly, VideoPrism had not been open-sourced. We present the results of using the more powerful open-source video encoder InternVideo2-1B [1] in Table 10 of Appendix A.3.
>
>
>
> [1] InternVideo2: Scaling Foundation Models for Multimodal Video Understanding

---

> > ### Author Response · Authors · 2025-11-22
> >
> > ## Q6. Why does the accuracy drop when increasing  from 64 to 128 in Figure 7 (a)? Intuitively, performance should improve as more frames are used to capture finer details, as suggested by the trend in Figure 7 (b).
> >
> > Our hypothesis regarding this phenomenon is that for extremely short videos (1–2 seconds), an excessively high sampling frame rate (128 frames) may not be beneficial for model learning. This is because an overly high frame rate can introduce a significant amount of redundant and repetitive information. For instance, a 2-second video recorded at 30 fps contains only 60 valid frames. If more frames are sampled, we can only compensate by duplicating existing frames to meet the required frame count. This duplication process may hinder the model’s ability to learn fine-grained temporal features.

---

### Official Review · Reviewer_rnXS · 2025-11-02

**Soundness:** 2
**Presentation:** 3
**Contribution:** 2
**Rating:** 4
**Confidence:** 3

**Summary:**

This paper presents VideoChat-Flash, a Multimodal Large Language Model (MLLM) designed to efficiently handle extremely long video contexts. The authors identify high computational cost and information redundancy as the primary bottlenecks in current long-video models. To address this, the paper introduces a comprehensive, four-part contribution: 1.A novel Hierarchical video token Compression method. This operates at two levels: (1) a Clip-level compression during encoding that uses a spatio-temporal video encoder to exploit local redundancy, and (2) a Video-level "progressive visual dropout" during LLM inference that prunes tokens based on attention patterns . 2.A new, large-scale dataset for long-video instruction tuning. 3.A multi-stage short-to-long learning scheme. 4.A new, more challenging "Needle in a Haystack" benchmark, the "Multi-Hop Needle-In-A-Video-Haystack".

**Strengths:**

1.The core strength is the HiCo compression framework. Achieving a 1/50 compression ratio (16 tokens/frame) while simultaneously achieving SOTA performance is a breakthrough for practical long-video MLLMs .The design is well-motivated.
2. The introduction of the Multi-Hop NIAH benchmark is a significant contribution.
3. The contributions of the paper are multifold including model design, data creation, and new evaluation.

**Weaknesses:**

1. The biggest concern is the novelty of the proposed method. While the system as a whole is novel and highly effective, its constituent parts are largely clever integrations of existing ideas.
2. The ablation in Table 2, while extensive, presents a slightly confusing narrative for HiCo. The baseline (196 tokens/frame) achieves a 63.7 on MLVU. The "+ HiCo" model (16 tokens/frame) drops to 60.6 on MLVU. This seems to contradict the "almost no performance loss" claim from the abstract. The performance is only recovered after applying the new training strategies and data. This suggests HiCo does incur a performance cost, which is then compensated for by the data and training recipe.

**Questions:**

1. Following on from my point in "Weaknesses," could you please elaborate on the performance drop when only HiCo is added (Table 2, row "+ HiCo")? How much of the final SOTA performance should be attributed to the sheer efficiency of HiCo (allowing for processing long contexts) versus the new LongVid dataset and short-to-long training strategy (which seem to compensate for an initial compression-induced quality loss)?
2. The authors mention that the video-level "Progressive Visual Dropout" is used "only during inference" due to "challenges in compatibility with training acceleration". Would there be issues such as training-inference mismatch ? However the results show this method is helpful. Could the author explain a bit why such mismatch does not lead to performance drop ?

---

> ### Author Response · Authors · 2025-11-22
>
> ## Q1:  While the system as a whole is novel and highly effective, its constituent parts are largely clever integrations of existing ideas.
>
> We sincerely thank the reviewer for recognizing the innovation and contributions of our system. A critical distinction between our work and prior art lies in our objective: **we pursue efficient long-video modeling to build long-video MLLMs from scratch**, whereas most existing works aim to accelerate existing MLLMs (e.g., by freezing most modules or adopting training-free approaches).
>
> We believe that learning native compressed representations for long-video modeling is essential. To this end, we propose the **Hierarchical Compression (HiCo)** paradigm. This is an **effective system-level design rather than merely a specific component**. HiCo decouples compression into two stages: handling visual redundancy at the **Clip-level** (within the ViT) and textual semantics at the **Video-level** (within the LLM). Furthermore, regarding specific design details, our approach offers key differences and novel insights compared to previous work:
>
> - **Clip-level Compression:** We propose leveraging spatio-temporal attention to capture local inter-frame redundancy. This allows the Video Encoder to **learn how to compress representations via parameter updates**, rather than relying on hand-crafted compression modules. This self-adaptive mechanism is key to achieving our exceptionally high compression rate.
> - **Video-level Compression:**  We introduce a new improved visual drop scheme specifically optimized for the long-video understanding domain.
> - **Timestamp Prompts:** We demonstrate that concise text prompts remain highly effective even under extreme visual compression. This avoids the computational overhead associated with adding text timestamps to every frame, a common practice in prior work.
> - **Duration-based Sampling:** We validate a more effective sampling strategy: applying dense sampling for short videos and sparse sampling for long videos. This differs from previous works, which predominantly utilize fixed FPS sampling or rely exclusively on sparse sampling.
>
> In summary, by combining these improvements, we are the first to demonstrate that **heavy-compression-based methods can achieve long-video understanding performance comparable to, or even surpassing, closed-source context-extension-based models.**
>
> ## Q2: This suggests HiCo does incur a performance cost, which is then compensated for by the data and training recipe. How much of the final SOTA performance should be attributed to the sheer efficiency of HiCo (allowing for processing long contexts)
>
> First, regarding the performance variations, while we acknowledge the slight decrease on MLVU (63.7 to 60.6) as noted, we actually observe performance gains on other benchmarks, such as MVBench (60.2 to 61.1) and VideoMME (52.8 to 53.2). Consequently, we consider the overall performance impact to be marginal. We believe this **minor trade-off is highly justifiable** given the substantial efficiency improvements HiCo provides.
>
> Second, we emphasize that the advantages of our compression strategy are twofold, benefiting both efficiency and performance:
>
> 1. **From an efficiency perspective:** With the same number of input frames, our method significantly reduces computational costs without a marked decline in performance, as demonstrated in the second row of Table 2.
> 2. **From a performance perspective:** In scenarios where computational resources and LLM context length are constrained—a common bottleneck in long-video understanding where standard sparse sampling often leads to severe information loss—compression allows the model to ingest a much larger number of video frames. This enables the model to achieve better performance by scaling up the input video context. To further empirically validate this, we present the additional data below. As shown in the table, under a fixed budget of total Vision Tokens (4,096), compressing the per-frame representation (from 256 to 64 tokens) allows us to significantly increase the number of input frames (from 16 to 64). This enhanced temporal context leads to substantial performance gains:
>
> | **Token per frame** | **Input Frames** | **Vision Token** | **VideoMME** | **MotionBench** |
> | ------------------- | ---------------- | ---------------- | ------------ | --------------- |
> | 256                 | 16               | 4096             | 51.3         | 44.8            |
> | 64                  | 64               | 4096             | 59.3(+8.0)        | 46.2(+1.4)            |

---

> > ### Author Response · Authors · 2025-11-22
> >
> > ## Q3: Could the author explain a bit why such mismatch does not lead to performance drop?
> >
> > We appreciate the reviewer for pointing this out and apologize for the confusion caused by our initial phrasing. **"Challenges in compatibility with training acceleration" was not the primary reason** for excluding Progressive Visual Dropout from the training phase.
> >
> > In our preliminary exploration, we were indeed concerned that the mismatch between training and inference (i.e., training with dropout vs. evaluating without) might negatively impact performance. Therefore, we conducted ablation studies and found that it did not yield significant benefits. In fact, incorporating it during training even resulted in a slight performance degradation:
> >
> > | Training                   | Evaluation                 | VideoMME   | MVBench    |
> > | -------------------------- | -------------------------- | ---------- | ---------- |
> > | No Dropout                 | No Dropout                 | 53.2       | 61.1       |
> > | Progressive Visual Dropout | No Dropout                 | 53.6(+0.4) | 58.4(-2.7) |
> > |                            | Progressive Visual Dropout | 52.2(-1.0) | 57.9(-3.3) |
> >
> > We hypothesize that this degradation stems from the underlying mechanism of Visual Drop, which relies on **text-guided attention** to identify and retain key visual tokens.
> >
> > - **The Premise:** Effective dropping requires the model to accurately determine which visual tokens are semantically relevant to the text.
> > - **The Conflict:** At the early stages of training, the MLLM has not yet established robust alignment between textual and visual modalities. Consequently, it cannot reliably identify "key" visual information. Forcing the model to drop tokens based on unlearned or unstable attention weights prevents it from effectively learning the full video representation.
> >
> > Therefore, based on these empirical results and analysis, we determined that applying Visual Drop exclusively during the inference stage.

---

> ### Comment · Reviewer_rnXS · 2025-11-28
> **reply**
>
> Thanks author for their detailed explanations. Most of my concerns are well addressed, and I would like to raise my score to 6 (marginally above the acceptance threshold)

---

### Author Response · Authors · 2025-11-22

Thank you to all the reviewers for your valuable feedback. We have further improved our paper based on your suggestions and would like to highlight several **key updates**:

- Regarding the impact of visual compression on efficiency and performance, **the novelity of our compression strategy**, and comparisons with other works, we have provided a more detailed discussion in Appendix A.3.
- Concerning **training and inference efficiency**, we have included a more thorough analysis in Appendix A.3.
- For Multi-Hop NIAH, we have included evaluation results using Gemini 2.5 Flash in Appendix A.6. We observed that **enabling the "thinking" mode significantly improves performance on the Multi-Hop NIAH task**, which we believe offers deeper insights into our benchmark.

We sincerely appreciate the reviewers’ valuable suggestions and hope to receive further feedback to continue refining our work!

---

### Meta-Review · Area_Chair_V37P · 2026-01-08

**Summary:**

This paper received 4 reviews. The reviewers (score/confidence) are: `m8kd (8/3), cNh3 (8/4), TNAS (6/5), rnXS (4/3)`. Their major concerns:

Methodology:
- The novelty of the proposed HiCo method is questionable -- its components are mostly integrations of existing ideas `rnXS (4/3)`.

Experiments:
- The ablation shows a performance drop (MLVU from 63.7 to 60.6) when only HiCo is added, contradicting the "almost no performance loss" claim; the attribution of final SOTA performance to HiCo vs. dataset/training strategy is unclear `rnXS (4/3)`.

**Reviewer Concerns:**

There is only one negative reviewer `rnXS (4/3)`, and the reviewer acknowledged that most of the concerns have been well addressed.

**Reviewer Scores:**

The 4 reviewers (score/confidence) are: `m8kd (8/3), cNh3 (8/4), TNAS (6/5), rnXS (4/3)`. Most reviewers are positive. The only negative reviewer also mentioned in the rebuttal: "Most of my concerns are well addressed, and I would like to raise my score to 6 (marginally above the acceptance threshold)".

---

### Decision · Program_Chairs · 2026-01-26

Accept (Poster)